# *Mycoplasma penetrans* methionyl-tRNA synthetase dimerizes via tandem N-terminal ancillary domains

**Behrouz Ghazi Esfahani[1,2,3], Madelynn K. Bowman[4], Nidhi Walia[1,5], Rebecca W. Alexander[4]\*, M. Elizabeth Stroupe[1]\***

1 Department of Biological Science, Institute of Molecular Biophysics, Florida State University, Tallahassee, Florida, United States of America, 2 Department of Biomedical Sciences, College of Medicine, Florida State University, Tallahassee, Florida, United Stated of America, 3 Department of Molecular Biosciences, Northwestern University, Evanston, Illinois, United States of America, 4 Department of Chemistry and Center for Molecular Signaling, Wake Forest University, Winston-Salem, North Carolina, United States of America, 5 Department of Biochemistry, Purdue University, West Lafayette, Illinois, United States of America

\* alexanr@wfu.edu (RWA); mestroupe@bio.fsu.edu (MES)

**Data availability statement:** The refined coordinates were deposited in the Protein Data Bank as 9OS7. The cryo-EM map and half maps were deposited in the Electron Microscopy Data Bank under the code EMD-70794. The micrographs

## Abstract

Diverse aminoacyl-tRNA synthetase gene fusions are now recognized as a common mechanism for enhancing genetic diversity across all domains of life. The *metS* gene from *Mycoplasma penetrans* is a striking example of such an evolutionary mechanism because although *M. penetrans* has a condensed genome, the *metS* gene is nearly twice the size of a typical bacterial gene encoding methionyl-tRNA synthetase (MetRS). We used cryo-EM to analyze the structure of the *metS* gene product (MpMetRS) to show that it is the fusion of three distinct enzyme domains: an N-terminal domain of unknown function, a dimeric alanine-glyoxylate aminotransferase (AGAT), and a MetRS. Only the first two N-terminal domains show two-fold symmetry and were resolved to 3.27 Å resolution; the MetRS domain is only partially resolved to 3.66 Å resolution. Modelling the full structure shows that a rotation of the MetRS domain relative to the AGAT domain must occur to accommodate a tRNA-bound MetRS. Further rearrangement of the catalytic domains would also be necessary to bring the active sites adjacent to one another if this unique assembly of catalytic domains functions to channel substrates to MetRS.

## Introduction

*Mycoplasma penetrans* is an obligate intracellular human pathogen that typically infects the respiratory or urogenital tracts of HIV-infected individuals [1]. Mycoplasma genomes are considerably smaller than those of other bacterial pathogens, challenging therapeutic targeting of canonical antibiotic pathways. Specifically, the *M. penetrans* genome contains open reading frames for 1038 proteins, compared to 4288

were deposited in the EMPIAR Data Base under the code EMPIAR-13467.

**Funding:** This study was supported by the National Institute of General Medical Sciences (RR024564), the National Science Foundation (MRI2017869) awarded to M.E.S, the National Institute of General Medical Sciences (GM145964), the National Institute of General Medical Sciences (RR025080), the National Institute of General Medical Sciences (GM139616), the National Science Foundation (MCB1856502) awarded to M.E.S., the National Science Foundation (CHE1904612) awarded to M.E.S. M.K.B. was supported by a Center for Molecular Signaling graduate fellowships. The funders had no role in study design, data collection and analysis, decision to publish, or preparation of the manuscript.

**Competing interests:** The authors have declared that no competing interests exist.

in *Escherichia coli*, so it relies on its host for many of its basic housekeeping and metabolic needs [2].

Despite *M. penetrans*' minimal genome, the *metS* gene that includes a region with homology to methionyl-tRNA synthetase (MetRS) is nearly twice the size of most bacterial MetRS enzymes [3]. Additional sequence homology predicts that the resulting MpMetRS gene product contains a class V pyridoxal phosphate-dependent aspartate aminotransferase domain that most closely aligns with alanine-glyoxylate aminotransferase (AGAT) just upstream of the tRNA aminoacylation domain [3]. In addition to the AGAT and MetRS domains, there is an N-terminal domain (NTD) that harbors ambiguous sequence homology, so its function is unknown. Together, MpMetRS has the following domain organization: NTD of unknown function – AGAT – MetRS (Fig 1A and Table 1). To date, no other organism with a similar chimeric MetRS has been identified, thus MpMetRS presents a unique drug target.

Many aminoacyl-tRNA synthetases (ARSs) have acquired polypeptide domains that extend function beyond the canonical tRNA aminoacylation activity [4]. These additional functions include transcriptional regulation, histidine biosynthesis, DNA binding, and mitochondrial RNA splicing [5]. Some MetRS enzymes contain a C-terminal domain that promotes dimerization and enhanced tRNA binding [6]. Eukaryotic MetRSs are typically present in the multisynthetase complex (MSC); noncatalytic domains appended to MetRS and other MSC components drive this assembly [7].

The *M. penetrans* AGAT and MetRS domains each perform their predicted enzyme function when expressed as the full-length enzyme or when the domains are expressed in isolation: the AGAT domain catalyzes the transamination of 2-keto-4-methylthiobutytrate (KMTB) to methionine (Met), amongst other aminotransferase reactions, and the MetRS domain aminoacylates tRNA^Met [3]. These two activities are functionally independent: mutation of essential residues in one domain does not impair activity of the other domain [3]. The isolated MetRS domain can selectively aminoacylate its cognate tRNA^Met and reject the near-cognate tRNA^Ile by contacting acceptor stem nucleotides remote from the tRNA anticodon [8]. Further, Met synthesized from KMTB by the AGAT domain can be attached to the 3'-end of tRNA^Met by the MetRS domain in the absence of exogenous Met [3]. How those domains are situated is unknown because there is no known structure of this first-to-be identified chimeric enzyme.

ARSs from diverse taxa have been biochemically dissected to explain their ancient and essential role in faithfully translating the genetic code. Many ARSs exhibit induced fit, both to select their cognate tRNAs based on nucleotide identities in the three-nucleotide anticodon and the acceptor stem and to discriminate amongst similar amino acid functional groups. Nevertheless, high-resolution structure determination techniques rely on specimen homogeneity, so many of the existing ARS structures have missing features that do not fully define the structural basis for the ARS mechanism. For example, tRNAs that are aminoacylated by class I sub-family ARSs necessarily undergo a "hairpin" rearrangement of the 3'-terminal acceptor end to approach the activated amino acid in the ARS catalytic site. Few high-resolution structures of such hairpinned tRNAs bound to their cognate ARSs are available. The

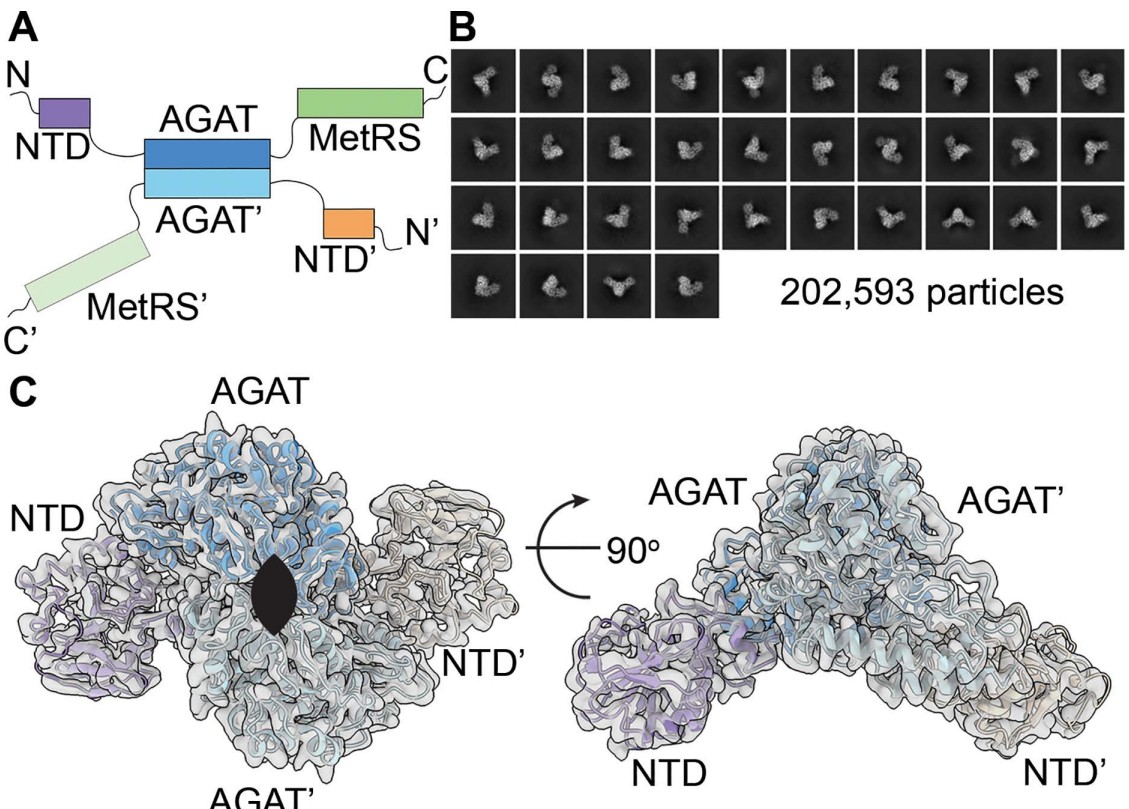

**Fig 1. MpMetRS domain structure and central dimer. A.** The *M. penetrans metS* gene product has the following domain structure from N-terminus (N) to C-terminus (C): NTD of unknown function – AGAT – MetRS and dimerizes through the central AGAT domain. **B.** Representative class averages showing high-resolution features and views with C2 symmetry. **C.** The first two domains form a C2 dimer composed of the NTD (purple or orange) and AGAT (azure or light blue) that is resolved to 3.27 Å-resolution. The C2 symmetry axis is perpendicular to the plane of the page in the left panel (flattened ellipse, black) and parallel to the plane of the page in the right panel.

**Table 1. Nomenclature.**

| Abbreviation | Usage |
|---|---|
| *metS* | gene encoding *Mycoplasma penetrans* MetRS |
| MpMetRS | full-length *M. penetrans metS* gene product |
| MetRS | general MetRS domain |
| AGAT | alanine-glyoxylate aminotransferase domain |
| NTD | N-terminal domain |
| ARS | general aminoacyl-tRNA synthetase |
| NTase | general nucleotidyl transferase |

modular nature of ARSs contributes to efficient catalysis and a wide array of novel functions, but a complete model of the structural basis for the biochemical data remains elusive. In the case of MpMetRS, ambiguity in sequence homology of the NTD of unknown function necessitated further experimental analysis.

Here, we report the 3.66 Å-resolution structure of MpMetRS that reveals a chimera of three structurally independent protein modules: 1) a previously uncharacterized NTD, 2) a dimeric AGAT domain, and 3) a single helical bundle from

one of the two MetRS domains. Structural homology of the NTD fold most closely aligns with a large class of nucleotidyl transferases (NTases), though its function and substrate specificity remain speculative. This NTD and the central ARS domain organize around the AGAT-dominated two-fold symmetry axis independently such that only the NTD and AGAT domain follow C2 symmetry. This is the first example, to our knowledge, of this collection of domains in a MetRS, allowing speculation that the three activities, nucleotide metabolism, methionine biosynthesis, and tRNA aminoacylation, may be functionally coupled in this minimal-genome organism.

## Materials and methods

### Preparation of biological specimens

Full length *metS* (GenBank MYPE9380) harboring a mutated codon for the M568A alteration to remove an internal start codon was cloned into pQE70 behind an N-terminal six-histidine tag [3,8]. The resulting plasmid was transformed into XL-10 Gold *E. coli* (Agilent, Santa Clara, CA, USA). Cells were grown at 37 °C in Luria Bertani broth with ampicillin selection until O. D.$_{600}$ = 0.6, the temperature was dropped to 25 °C, and then induced with 1 mM IPTG. Cells were further grown overnight at 25 °C before being harvested by centrifugation at 4,000 x g for 30 min. Cells were resuspended in the standard buffer: 50 mM HEPES/pH 6.8, 150 mM KCl, and 10 mM imidazole.

For purification, protease inhibitors (Roche Diagnostics, USA), 1 mM PMSF, and 1 mM pepstatin-A were added, followed by lysis with a microfluidizer and clarification by centrifugation at 13,000 x g for 30 min. Supernatant was then applied to an Ni-NTA affinity column (Cytiva, Marlborough, MA, USA) in the lysis buffer, washed with 50 mM HEPES/6.8, 500 mM KCl, and 10 mM imidazole to remove contaminating nucleic acid. After re-equilibration in the loading buffer, the sample was eluted with 250 mM imidazole and dialyzed into 50 mM HEPES/6.8, 150 mM KCl, and 10 mM imidazole. Next, the sample was loaded onto a heparin affinity column (Cytiva) to remove any residual nucleic acid-bound protein. Gradient elution to 500 mM KCl followed, where MpMetRS eluted at about 200 mM KCl with an A260:280 ratio of 0.5. Sample was polished over a Sephacryl S300 (Cytiva) size exclusion chromatography (SEC) column and then concentrated to ~10 mg/mL. Blue Native PAGE analysis (ThermoScientific, Waltham, MA, USA) was performed on protein at 0.02 mg/mL [9]. Analytical SEC was performed with a Superose 6 column (Cytiva) on a sample at 1 mg/mL. The Superose 6 column was calibrated using a High Molecular Weight (MW) Calibration Kit (Cytiva) according to the manufacturer's protocol after running blue dextran to identify the void volume (8.1 mL). The known standards were normalized for the void volume ($K_{av}$), then plotted against the log of their MWs to determine the relationship between running volume and MW. That relationship (log(MW) = −0.31($K_{av}$) + 1.1) was used to determine the apparent MW of MpMetRS.

*Mycoplasma penetrans* tRNA$^{Met}$ (MptRNA$^{Met}$) was prepared by *in vitro* transcription from overlapping oligonucleotides as described by Sherlin and coworkers [10] using sequence information from the Genomic tRNA database [11]. Transcript was purified by 8M Urea-PAGE followed by elution in 0.5 M ammonium acetate (pH 5.3), 1 mM EDTA and ethanol precipitation at −20 °C. tRNA was resuspended in TE buffer; concentration was determined by UV/visible absorbance at 260 nm using the extinction coefficient 40 (μg/mL)$^{-1}$ cm$^{-1}$.

### Cryo-EM sample preparation

MpMetRS adheres to the graphene surface, allowing low concentration to avoid protein aggregation. Therefore, MpMetRS (4 μL of 0.04 mg/mL or 0.04 mg/mL MpMetRS with 5-fold molar excess folded tRNA$^{Met}$, in the absence of the aminoacyl substrate), was applied to a hydrophilized graphene-coated [12] holey carbon-on-gold Quantifoil cryo-EM grid (Quantifoil, Jena, Germany). Plunging into liquified ethane after a blot force of 4, 1 s blot was performed on a Vitrobot Mark IV (Thermo Scientific), resulting in thin ice and an even distribution of monodispersed particles. Alternatively, 10 mg/mL MpMetRS or MpMetRS with 5-fold molar excess tRNA$^{Met}$ was applied to a nanowire self-wicking grid using the chameleon® (SPT Labtech, Melbourne, UK) nanospray device [13] with a 54 ms wicking time.

## Cryo-EM data collection and processing

Images were collected using a Titan Krios transmission electron microscope (Thermo Fisher Scientific, Waltham, MA, USA) operating at 300 kV on a K3 camera (Gatan, Waltham, MA, USA) with the Leginon automated data acquisition package [14], housed in the Florida State University Biological Science Imaging Resource. For apo-MpMetRS, a total of 4,000 movies were collected with a pixel size of 0.86 Å/pixel. For MpMetRS with tRNA$^{Met}$, 4,400 movies were collected at 1.12 Å/pixel. After motion correction using MotionCor2 [15] in the Relion-3 GUI [16], CTF estimation was performed in CTFFIND4 [17] for both datasets. Particles were initially selected using the "blob picker" algorithm, analyzed with 2D classification to identify appropriate templates, then refined with the "template picker" in CryoSPARC [18]. All subsequent analysis was performed in CryoSPARC.

Further refinement of the apo-MpMetRS dataset by template-free 2D classification identified 202,593 particles with clear 2-fold symmetry that were used for initial model building. Non-uniform 3D refinement with imposed C2 symmetry resulted in a 3.27 Å-resolution map. The map showed no evidence of the MetRS catalytic domain. Closer inspection of the 2D classes revealed some classes with blurred features that did not follow the 2-fold symmetry that dominated the initial classification (S1 Fig in S1 File). Further, the GSFSC curve of the C2 dimer showed a bump at moderate resolution, indicative of non-resolved features in the map (S2 Fig in S1 File) To explore the possibility that this density corresponded to the missing MetRS catalytic domain, this limited subset of particles (60,000) was used to refine a non-uniform, asymmetric map to 3.66 Å resolution. All maps were sharpened with deepEMhancer for enhanced visualization [19].

For the MpMetRS/tRNA$^{Met}$ specimen, 424,722 particles were used for 2D analysis and refined using the same protocol. Further 3D analysis revealed a similar structure of equivalent resolution to that of apo MpMetRS.

## Model building and refinement

Model building was initiated with the ModelAngelo [20] model builder. Iterative real-space refinement in PHENIX [21], with manual fitting in Coot [22], was performed independently for the core dimer and the C-terminal domain helical bundle of the MetRS domain (S1 Table in S1 File). To identify the nature of the NTD, the atomic coordinates for the isolated domain were submitted to the DALI server [23]. Likewise, the atomic coordinates for the isolated AGAT domain were submitted to the DALI server. Further structural analysis was performed in ChimeraX [24].

## Dynamic light scattering

To better understand the oligomeric state of the complex, we used dynamic light scattering (DLS) to assess the apparent MW and hydrodynamic radius ($R_h$) of the complex at 5 mg/mL. Protein was diluted in standard buffer and centrifuged for 10 m at 14,000 x g. Measurements were made on a Dyna Pro-99 instrument along with a DynaPro-MSXTC temperature-controlled microsampler (Wyatt/Protein Solutions, Santa Barbara, CA, USA). Data were analyzed in the Dynamic v7.0.0.94 software. The theoretical $R_h$ for the model of the full dimer was calculated using the HullRad server [25].

## Results

### The NTD has structural homology to an NTase

Initial 2D analysis shows classes with high-resolution features that correspond to the dimer mediated by the AGAT domain (Fig 1B and S1 Fig in S1 File). Refinement and 3D analysis of all 202,593 particles further supports this observation, showing only density for the dimeric core that resolves to 3.27 Å resolution (Fig 1C and S1 and S2 Figs in S1 File).

A DALI search on the previously uncharacterized NTD showed that out of the top 34 non-redundant structures with z-score ≥10, 26 (76%) are NTases. The highest z-score belongs to an NTase (PntC) from *Treponema denticola* [26] (Table 2). Others are identified as NTases with specificity to CTP or UTP. Three-dimensional alignment of six non-redundant top hits

**Table 2. Exemplars from structure-based homology search of the NTD.**

| PDB | Organism | Nucleotide | RMSD with the NTD |
|---|---|---|---|
| 6PD1 | *Treponema denticola* | CTP | 1.2 Å (190 atom pairs) |
| 5HS2 | *Bacillus subtilis* | CTP | 1.1 Å (38 atom pairs) |
| 2WE9 | *Mycobacterium tuberculosis* | unknown | 1.3 Å (53 atom pairs) |
| 9MH4 | *Klebsiella aerogenes* | UTP | 1.3 Å (45 atom pairs) |
| 2XMH | *Archaeoglobus fulgidus* | CTP | 1.2 Å (39 atom pairs) |
| 7D73 | *Homo sapiens* | GTP | 1.1 Å (33 atom pairs) |

identifies the domain's structural features common to NTases but also explains the low sequence homology to other members of this large enzyme family (Fig 2A and S3A Fig in S1 File).

Specifically, the MpMetRS NTD shares common core structural elements with NTase family members: a central three-stranded mixed β sheet flanked by two α helices on either face (S3B Fig in S1 File) [29]. β-strands 1, 4, and 6 form the core and an additional four strands (2, 3, 5, and 7) fill out a seven stranded β-sheet. Strand 6 is the only antiparallel strand in the mixed β-sheet. A total of six α-helices, with α-helices 2 and 5 on the bottom face of the β-sheet and α-helices 4 and 6 on the top, complete the core structure. A central cavity at the edge of the β-sheet is in a similar position to the nucleotide binding pocket in a canonical family member (Fig 2A).

The loop that caps the nucleotide binding cavity is another structural element that differentiates the NTD fold from the core NTase domain (residues 18–25, Fig 2B and S3A Fig in S1 File). Of the six other structures identified by DALI, only one was bound to its nucleotide substrate, the 2-C-Methyl-D-erythritol-4-phosphate cytidyltransferase (IspD) from *Bacillus subtilis* [27]. The others were in their apo form, as is the structure determined here. Nevertheless, in each of those structures, the loop that surrounds the nucleotide-binding cavity is in an opened conformation to make space for the nucleotide. In contrast, the same loop in

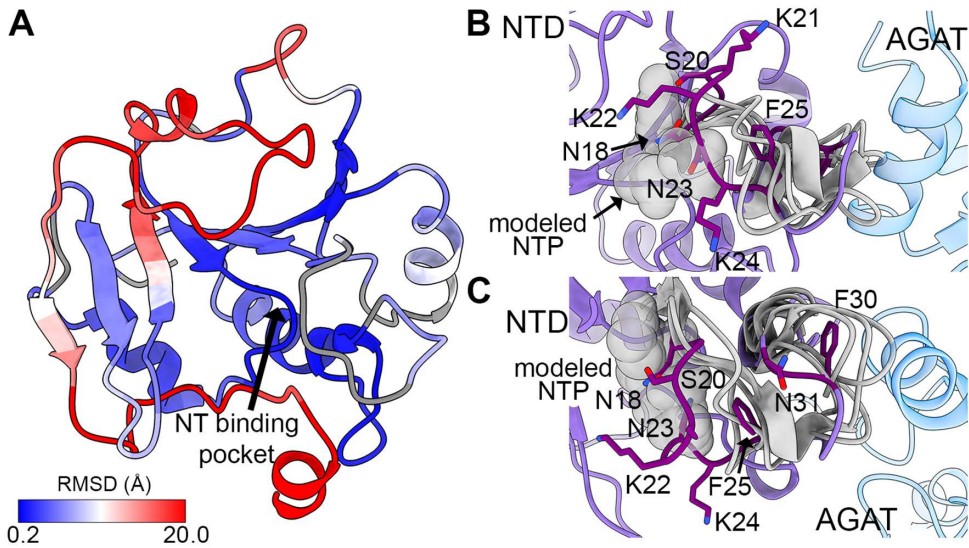

**Fig 2. Structural properties of the NTD. A**. Structural alignment shows a conserved NTase-like fold with a central mixed three-stranded β-sheet between α-helices but with additional structural motifs. This structurally conserved Rossmann fold is central to the domain (0.2 Å RMSD, blue) whereas the divergent motifs are at the periphery (20 Å RMSD, red). Non-conserved regions are gray. **B.** A structurally divergent loop (dark purple) from the NTD (purple) encloses the pocket where a nucleotide (modeled in light gray from PDB ID 5HS2 [27]) binds in other NTases (gray, PDB IDs 6PD1 [26], 9WE9, 9MH4, and 2XMH [28]). **C.** Another structurally divergent loop (dark purple) moves toward the NTD, away from the MpMetRS AGAT domain, in contrast to the other structural homologs shown as in **(B)**.

MpMetRS is longer and is closed, sterically obscuring the pocket and clashing with a potential nucleotide binding site. After a turn, residues 29–31 also form a loop before the first peripheral α-helix that, in MpMetRS, packs against the AGAT domain (Fig 2C and S3 Fig in S1 File). In the other homologs analyzed here, this loop is free and further from the nucleotide binding pocket. In this way, access to the nucleotide binding pocket could be related to the position of the subsequent AGAT domain.

## The central domain is a dimeric AGAT

A *de novo* model of the central domain supports the prediction that it is a class V pyridoxal phosphate (PLP)-dependent aspartate aminotransferase superfamily member, homologous to the AGAT from *Anabaena* sp (Fig 3A and S4 Fig in

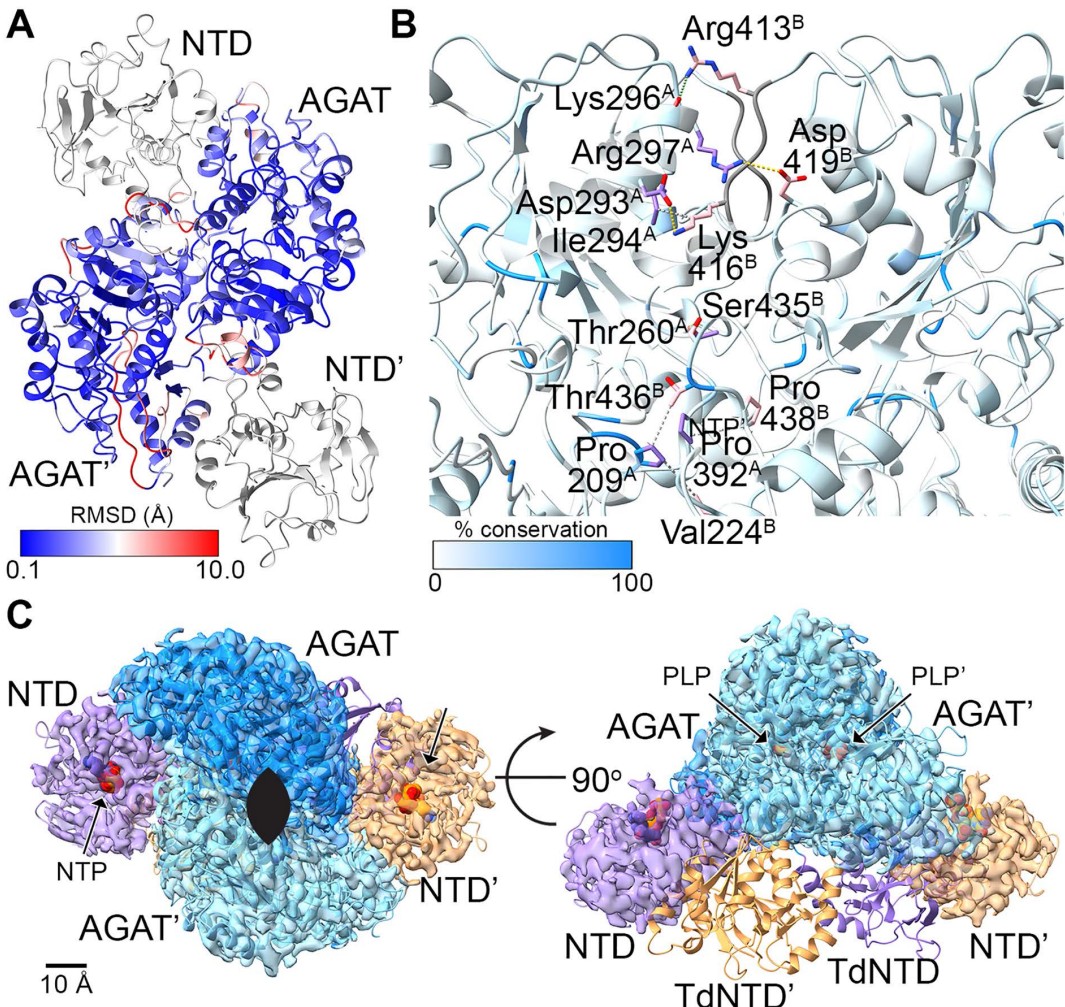

**Fig 3. Structural properties of the central AGAT domain. A.** Structural alignment shows a conserved AGAT (0.1 Å RMSD, blue) with minimal divergent elements (10 Å RMSD, red). The NTD is light gray for reference. **B.** The dimer interface is structurally conserved across homologs, but sequence conservation is low (the backbone is colored white-to-azure blue: 0% conservation is white and 100% conservation is azure blue). Backbone from sequence gaps is gray. Sidechains from one subunit are purple and from the C2 symmetric subunit are pink. Hydrophobic interactions are depicted by gray dotted lines, hydrogen bonds are green dotted lines, and ionic interactions are yellow dotted lines) **C.** The NTD from MpMetRS (purple or orange cryo-EM map) is positioned in a different orientation relative to the AGAT dimerization domain than it is in a similar chimeric enzyme from *T. denticola* (purple or orange ribbon, PDB ID 6PD1 [26]). The putative NTP binding site is marked by an NTP modeled from PDB ID 5HS2 [27] and, along with the PLP cofactor, is shown as space-filling atoms. Scale bar = 10 Å. The C2 symmetry axis is marked by an elongated ellipse (black).

S1 File) [3,30]. Based on that prediction, the isolated domain has previously been shown biochemically to be an aminotransferase that can use various amine donors to synthesize different amino acids including Met [3]; the domain will be referred to here as an AGAT for simplicity. Other superfamily members identified through the structural homology search use diverse substrates, including phosphoserine, serine, 2-aminoethylphosphate (AEP), and ureidoglycine (Table 3).

In each homolog, the dimer interface buries about 3504 Å$^2$ (of a total of 23,060 Å$^2$ for each monomer including the NTD and AGAT) and is mediated by hydrophobic stacking interactions, salt bridges, and hydrogen bonds (Fig 3B). The interface is conserved amongst the structural homologs, but each has an idiomatic set of interactions that make up the interface. In MpMetRS, a core of hydrophobic interactions includes stacking interactions between Ile294Cγ1 and Lys416Cδ; two stacked prolines (Pro392 and Pro438); and an *en face* interaction between Thr436Cγ and Pro209, that subsequently interacts with ValCγ. Salt bridges include those between Arg297Nη and Asp419Oδ and between Asp293Oδ and Lys416 Nζ. Arg413Nη1 forms a hydrogen bond with the carbonyl from Lys296 and Thr260OH forms a hydrogen bond with Ser435OH across the interface. Of this set of interactions, Thr436 and Pro209 are the only conserved residues amongst those analyzed here.

The structure PDB ID 6PD1 [26] from *T. denticola* was identified as a top hit from both DALI searches because it is a fusion between PntC (a cytidylyltransferase) and AEPT (an AEP transaminase) with homology to the same NTase and aspartate aminotransferase superfamilies as in MpMetRS, where two AGAT-like domains dimerize with the C2-symmetric NTase-like domains at the periphery (Fig 3C) [26]. In MpMetRS, the AGAT active sites, marked by the PLP cofactor, sit about 30 Å apart from one another, 40 Å away from the predicted nucleotide binding pocket on the NTD from the same polypeptide and about 50 Å away from the nucleotide binding pocket of its partner. The predicted nucleotide binding pockets sit about 80 Å away from one another. In contrast, the active sites from each enzyme domain in the TdPntC-AEPT fusion are separated by about 50 Å within the same polypeptide as well as across the dimer. The nucleotide binding pockets are about 50 Å away from one another. Interestingly, the NTDs are swapped relative to their AGAT domains when compared to each other. That is, an AGAT following an NTD within the same subunit in MpMetRS are close, whereas the NTDs are closer to the AGAT from the opposite subunit in TdPntC-AEPT.

## The position of the MetRS domain does not follow C2 symmetry

To better understand the relationship between the NTD, AGAT, and MetRS domains, we performed further 3D analysis of a subset of particles that exhibit additional, less-well defined features in 2D (Fig 4A, S1 and S5 Figs, and S1 Table in S1 File). The resulting structure shows additional density with helical features, albeit at limited resolution (3.66 Å resolution) (Fig 4B). Modeling of the MetRS domain confirms homology to other similar class Ia ARSs where the C-terminal anticodon-binding domain forms an antiparallel five-helical bundle. There is no well-resolved density for the N-terminal catalytic site of MetRS or for MetRS from the opposing subunit, despite PAGE-analysis showing the purified full-length

**Table 3. Exemplars from structure-based homology search of the AGAT domain.**

| PDB | Organism | substrate | RMSD with the MpAGAT |
|---|---|---|---|
| 2FYF | *Mycobacterium tuberculosis* | phosphoserine | 1.3 Å (137 atom pairs) |
| 3KE3 | *Psychrobacter arcticum* | serine-pyruvate (putative) | 1.3 Å (158 atom pairs) |
| 6PD1 | *Treponema denticola* | 2-aminoethylphosphonate | 1.2 Å (191 atom pairs) |
| 6PK3 | *Arabidopsis thaliana* | alanine-glyoxylate | 1.3 Å (45 atom pairs) |
| 1VJO | *Anabaena sp* | alanine-glyoxylate | 1.3 Å (220 atom pairs) |
| 3NNK | *Klebsiella pneumoniae* | ureidoglycine | 1.3 Å (200 atom pairs) |

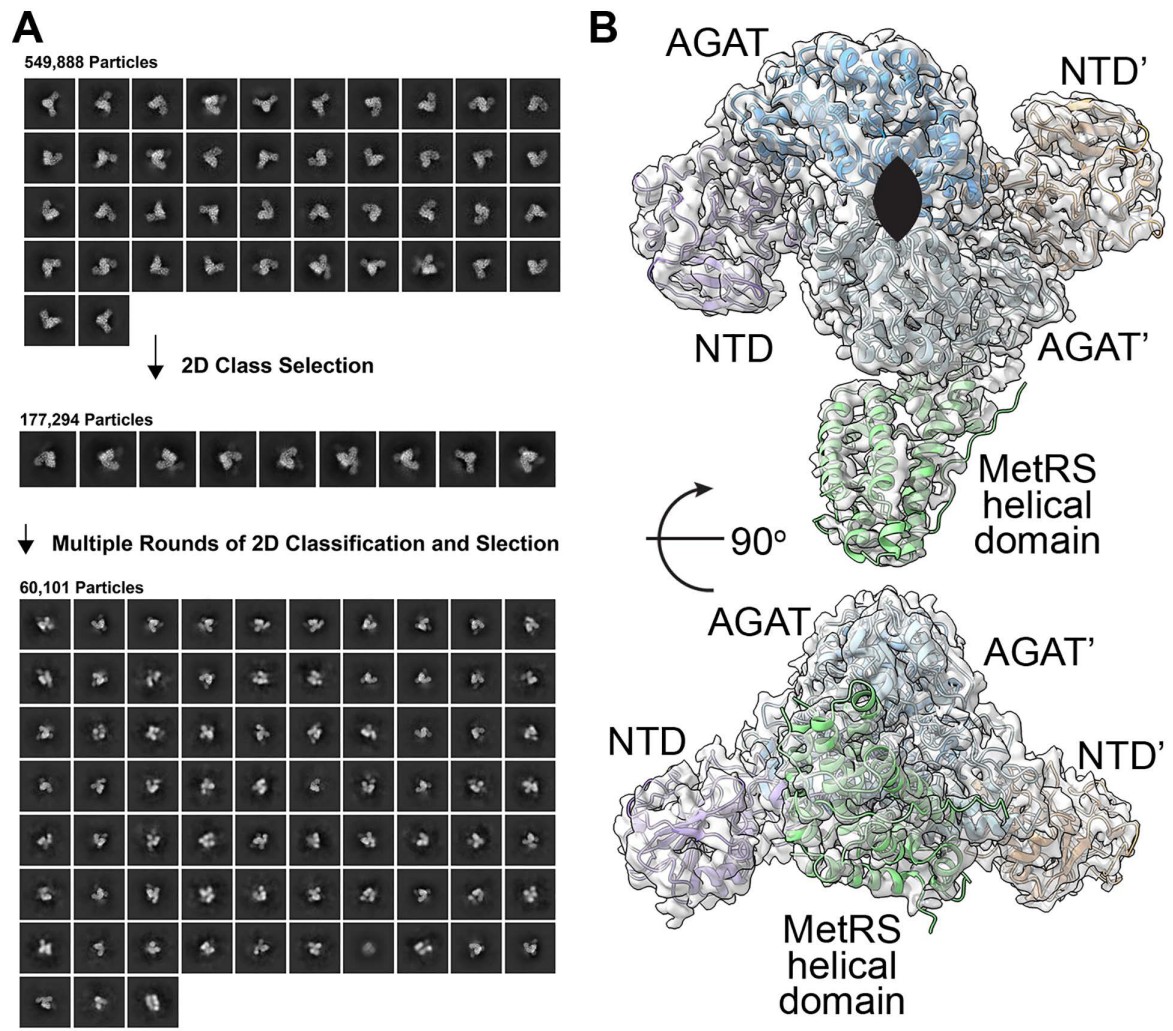

**Fig 4. Asymmetric analysis of MpMetRS. A.** 2D analysis of a subset of 177,294 with asymmetric features were subject to iterative 2D classification and visual inspection to identify 60,101 particles with defined density from a single MetRS helical domain. **B.** The MetRS helical domain (light green) does not follow C2 symmetry when compared to the NTD (purple or orange) and AGAT domain (azure or light blue).

protein that is predominantly a dimer (S6A Fig in S1 File), therefore we interpret the visible helical domain as coming from only one of the subunits and not including the catalytic domain.

When sample was pre-mixed with excess tRNA, new 2D classes appeared with additional, ill-defined density at the outside of the central, well-defined dimer regardless of the method used for plunging the specimen (40% compared to 0% without tRNA) (S7 Fig in S1 File). Despite extensive 3D classification, however, no high-resolution features appeared for the catalytic domain or bound tRNA. Identifying conditions where the tRNA binds for structural analysis remains an active area of exploration.

## Modelling

The loop that connects helix 2 and 3 of the MetRS helical bundle tucks into the AGAT domain in a crevice formed between two of its helices at the opposite face of the dimer interface (Fig 5A). Docking the helical bundle from monomeric *E. coli*

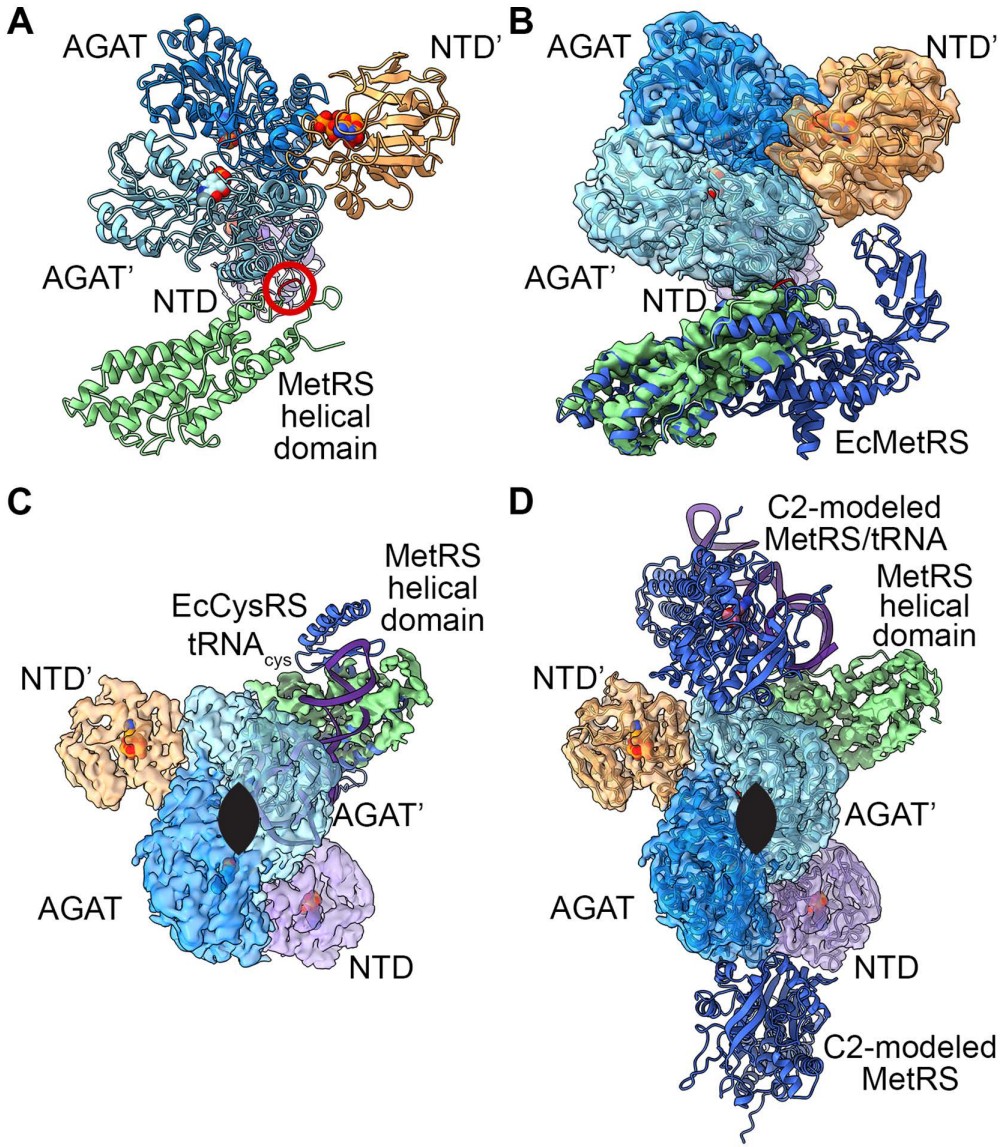

**Fig 5. The MpMetRS dimer does not accommodate a tRNA-bound ARS. A.** MpMetRS is colored as above with the putative nucleotide binding site and PLP marked by space-filling atoms. The turn that interacts with the AGAT is red. **B.** Docking the helical domain from apo EcMetRS (PDB ID 1QQT [31], dark blue) onto the MpMetRS helical domain positions the zinc-binding domain, marked by a gray sphere, between the NTDs. **C.** A bound tRNA (dark purple, from EcCysRS, PDB ID 1U0B [32]) would sterically conflict with the NTD-AGAT dimer if the MetRS domain were positioned as resolved in this structure. **D.** If the whole MpMetRS structure were to obey the same C2 symmetry as the NTD-AGAT domain, modeled in AlphaFold2 [34], the MetRS domain would fit such that the tRNA ligand would face away from the putative NTD nucleotide binding domain. The 3' end of the tRNA is also shown as space filling atoms.

MetRS (PDB ID 1QQT) [31] predicts that the catalytic domain would fall immediately between the C2-symmetric NTDs (Fig 5B) (RMSD of 1.0 Å for a core of 108 residues). In contrast, docking the tRNA-bound *E. coli* CysRS [32] onto the helical bundle of MpMetRS shows that the tRNA would sterically clash with the N-terminal dimer as positioned in the apo structure (Fig 5C) (RMSD of 1.2 Å for a core of 40 residues). The tRNA-bound CysRS co-crystal structure (PDB ID 1U0B [32]) was used as a template because it captures the hairpinned tRNA acceptor stem directed into the ARS active

site; a cocrystal structure of *A. aeolicus* MetRS:tRNA$^{Met}$ is available (PDB ID 2CSX [33]), however in this structure the tRNA 3'-end is disordered and the acceptor stem does not approach the active site. Despite our efforts to bind tRNA$^{Met}$ to the protein and the appearance of additional densities in the 2D averages (S7 Fig in S1 File), no additional density was observed in the reconstruction.

To better understand how this steric clash could resolve, we compared the experimental results, which show an asymmetric and incompletely ordered dimer, with those from AlphaFold2 [35], imposing C2 symmetry. To adopt the fully symmetric dimer predicted by AlphaFold2, the helical subdomain from the MetRS domain would need to rotate about 60° using the AGAT active site as a pivot point reference (Fig 5D). With that rotation, the MetRS catalytic domain is removed from the space between the two NTDs, opening the tRNA binding site (RMSD of 1.3 for a core of 111 residues between the MetRS domain and EcCysRS). The 3' end of the tRNA sits about 46 Å away from the AGAT active site.

### MpMetRS oligomeric state depends on the protein concentration

The above results were determined from a 0.04 mg/mL sample applied to graphene-coated grids to protect the protein from the air-water interface and concentrate it on the substrate. In contrast, the chameleon® plunging system (SPT Labtech) allows for sample application at a higher concentration, in this case ~10 mg/mL, while blot-free thinning and rapid freezing can avoid air-water interface complications [13]. 2D analysis of this preparation shows additional classes that appear with additional, well-resolved densities (Fig 6A). Further 3D analysis of these subsets of particles reveals primarily tetrameric NTD-AGAT assemblies (Fig 6B). Within those higher-order oligomers, the NTD and AGAT domains appear concatenated, with the NTD of one polypeptide positioned in opposition to the AGAT domain of the next polypeptide to bury an additional 1575 Å$^2$. No density for the MetRS domain was observed in these particles.

In solution and at low concentration (~0.5 mg/mL), as measured by SEC, MpMetRS primarily elutes as a peak that corresponds to a mass of 260 kDa, close to the theoretical mass of a dimer (S6B Fig and S2 Table in S1 File). A portion of the sample eluted as a smaller specimen (156 kDa), possibly from partial C-terminal degradation. Only the peak fraction that eluted at 14 mL was collected and concentrated for cryo-EM imaging. DLS on the same peak fraction that was concentrated to 5 mg/mL revealed a particle with a mass of 527 kDa – twice that of the theoretical dimer (S6C Fig in S1 File). A small portion of the protein was aggregated at this concentration. The theoretical $R_H$ at this concentration is 8.7 nm, twice the theoretical $R_H$ of full C2 symmetric MpMetRS generated by modelling the dimer containing all three domains (4.3 nm).

### Discussion

MpMetRS is included amongst a growing class of ARS enzymes where the aminoacylation domain is expressed as a chimera with a "helper" protein, often of housekeeping function [36]. The triple fusion of a putative NTase, AGAT, and ARS is, to date, unique. *M. penetrans* has a minimal genome, relying on the human host for many metabolic and regulatory functions. The structural relationship of these domains leads to questions as to why this assembly was evolutionarily favorable.

Structural, not sequence, homology of the NTD identifies its core as most closely resembling a Rossmann fold dinucleotide binding motif from a diverse superfamily of nucleotidyltransferases, including cytidylyl-, uridylyl- and guanylyltransferases. Both structural and sequence homology of the central domain confirm this motif as a class V pyridoxal phosphate-dependent aspartate aminotransferase. Another example of such a fusion comes from a spirochete, *T. denticola*, where it is believed to be involved in natural product synthesis of a phosphonate that is used for modifying its cell surface [26]. Structural and sequence homology of the central domain confirms prior biochemical analysis showing it is an AGAT-like aminotransferase that can use various amino donating groups to synthesize Met [3]. To our knowledge, this is the first example of that fusion of enzyme domains added to an ARS. The advantage for an organism with a minimal

**A** 3,468 Particles Out of 308,064 Total Particles

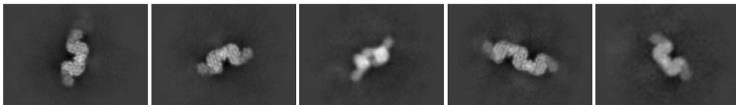

**B**

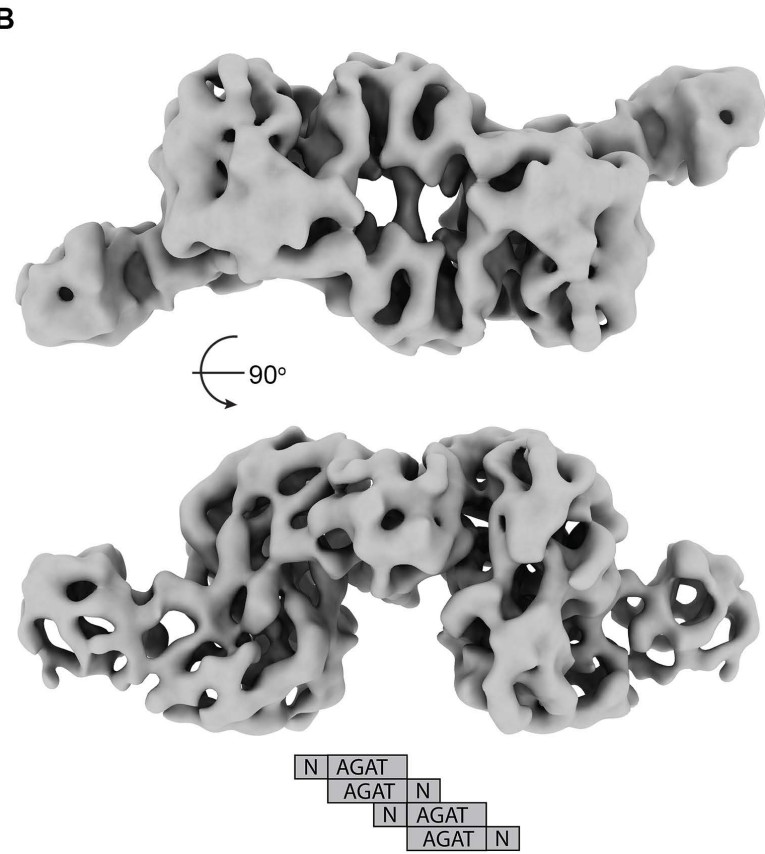

**Fig 6. At higher concentrations MpMetRS assembles into higher order structures. A.** 2D class averages with well-defined density from a subset of the total particles are twice the size of dimeric MpMetRS. **B.** The primary form of the concatenated molecules is a tetramer with an additional interface between an AGAT domain and the NTD from another dimer.

genome to evolve a MetRS with additional capabilities remains to be fully understood but allows some speculation that the activities could be coupled.

To further explore the possibility that the active sites could be coupled to increase the local concentration of Met for aminoacylating tRNA, we modeled the apo anticodon binding subdomain based on the crystal structure of EcCysRS [31] (Fig 5). This simple docking positions the zinc-binding subdomain from the MetRS between two NTDs, but on the opposite face from the putative NTase active site. Prior molecular dynamic simulations performed on a monomeric form of EcMetRS, which shares 25% sequence identity and 45% sequence similarity with the MetRS domain of MpMetRS, predicted that the zinc-binding insertion between halves of the active site, a surface loop that caps the active site, and the anticodon binding domain are particularly mobile and implicated in MetRS function [37]. This implied role of dynamics of those domains in tRNA aminoacylation also likely explains why these elements are not visualized in the averaged cryo-EM structure of MpMetRS.

When the same docking is performed using a tRNA-bound ARS, EcCysRS [32], a conflict arises from the tRNA binding pose relative to the cleft between the central, C2-symmetric NTD/AGAT domain. Specifically, the MetRS helix-turn-helix motif that interacts with the AGAT domain binds to the inside elbow of the tRNA when the substrate is bound, thus the apo conformation of the MetRS domain relative to the AGAT domain conflicts with the substrate-bound conformation (Fig 5C). We predict that a large conformational change occurs, perhaps through the linker between the core AGAT and MetRS domains, to accommodate the tRNA substrate and perform the aminoacylation of Met-tRNA. This linker region is disordered in apo MpMetRS such that it is not visible in the 3D reconstruction, despite extensive classification. For that reason, we cannot assess whether the visible helical domain, which is C-terminal to the aminoacylation sub-domain, originates from chain A or chain B.

We further observed a concentration-dependent effect on the oligomeric state of MpMetRS. Oligomerization of nucleotide and polynucleotide binding proteins is not atypical, for example in the DNA recombination protein RecA [38], cytidine triphosphate synthase [39] or the G-quadruplex binding protein nucleoside diphosphate kinase [40]. The *in vivo* significance of this observation for MpMetRS remains to be determined as we do not know of any specific other examples of concentration-dependent NTases or AGAT enzymes in isolation except for that of PntC-AEPT, described below. At high concentrations of MpMetRS, two NTDs dimerize orthogonally to the AGAT dimeric interface through a loop around the

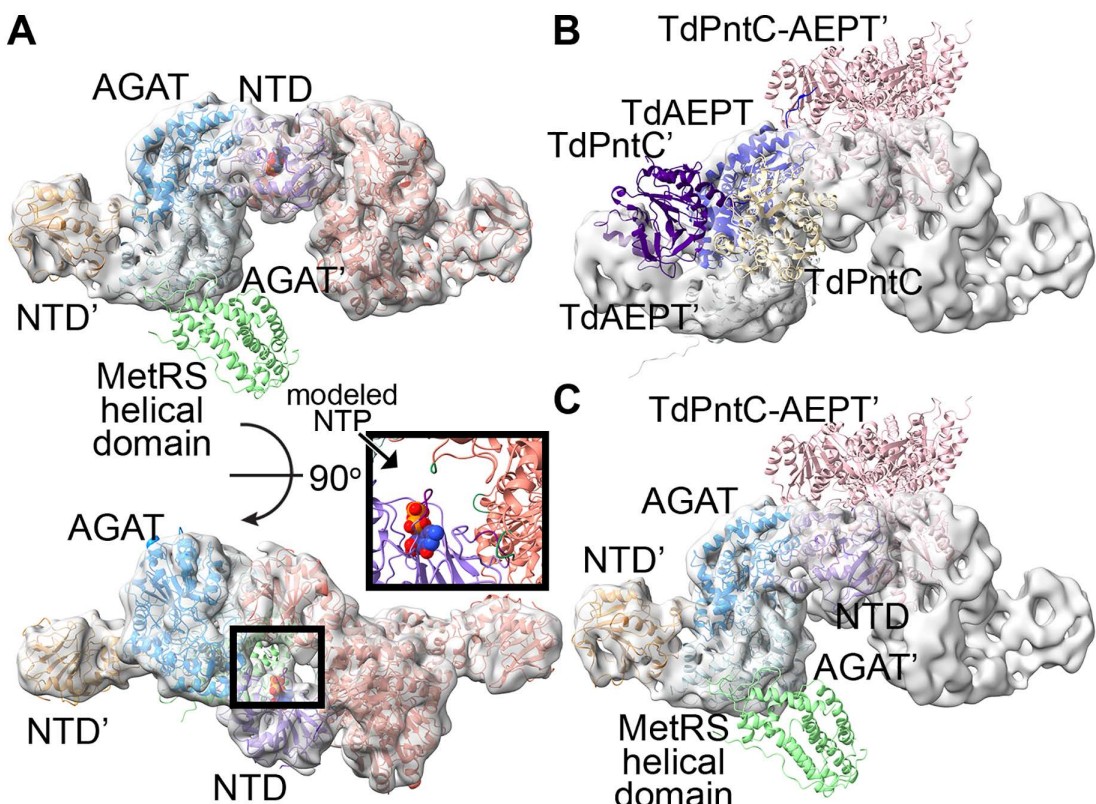

**Fig 7. MpMetRS forms higher-order complexes like TdPntC-AEPT. A.** Tetramerization of MpMetRS (the domains of the first dimer colored as described in Fig 4 and the second dimer in coral) does not occlude the position of the MetRS helical domain, but the MetRS domain is not resolved in the structure. **B.** TdPntC-AEPT (one subunit colored with the PntC domain in tan and the AEPT in light purple, the C2-symmetric subunit colored with the PntC in dark purple and the AEPT domain in medium purple) also forms a dimer of dimers (second dimer in pink) in the asymmetric unit of the crystal. The resulting tetramer is not the same as in MpMetRS (light gray) **C.** The TdPntC-AEPT dimer of dimers is positioned on the opposite face from the MetRS helical domain from MpMetRS.

predicted nucleotide binding site (Figs 5 and 7A). A second interaction between the NTD and AGAT domain from separate polypeptides forms while maintaining an open channel to the predicted nucleotide binding site (Fig 7A, inset), so presumably the substrate and product can still diffuse into and out of the binding pocket. The asymmetric unit of the TdPntC-AEPT also contains a dimer of dimers that buries 1154 $Å^2$ [26], but the second dimer is on a different face of the primary PntC-AEPT dimer from where it is in MpMetRS (Fig 7B). Although the MetRS domain is not sufficiently ordered to see its density, there is no steric conflict between its position in the dimer and the tetramer when the helical domain is positioned as it is in the asymmetric structure (Fig 7C). The first two N-terminal domains of MpMetRS define its oligomeric state, but in a different way than in TdPntC-AEPT. More examples need to be identified and analyzed to determine the rules that control assembly and/or biological function, but the determinants are likely species specific.

## Conclusions

Coordination of the three activities might explain the evolutionary advantage of having each activity co-localized within a single polypeptide, even in an organism that has a minimal genome. First, if the NTD serves as a NTase, it would increase the availability of the precious cellular resource ATP. Second, the AGAT domain serves as the core to the MpMetRS dimer and synthesizes methionine to possibly supply the aminoacylation catalytic domain. Finally, the mobile MetRS domain could position its catalytic site to receive these essential metabolites before binding tRNA$^{Met}$. Future enzymatic studies will evaluate the synergy that benefits *M. penetrans* from fusing these different enzyme activities.

## Supporting information

**S1 File. Supplemental figures and tables.**
(DOCX)

## Acknowledgments

We thank Dr. Christopher Stroupe and Jen Kennedy for helpful discussions. Florida State University supports cryo-EM in the Biological Imaging Resource Center, which houses the following equipment used in this study: a Gatan Solaris Plasma Cleaner, a Hitachi HT7800, a ThermoFisher Vitrobot Mark IV, an SPI chameleon® plunging system, a ThermoFisher Titan Krios, and a DE Apollo direct electron detector.

## Author contributions

**Conceptualization:** Rebecca W. Alexander, M. Elizabeth Stroupe.

**Data curation:** Behrouz Ghazi Esfahani.

**Formal analysis:** Behrouz Ghazi Esfahani, Nidhi Walia.

**Supervision:** Rebecca W. Alexander, M. Elizabeth Stroupe.

**Writing – original draft:** Rebecca W. Alexander, M. Elizabeth Stroupe.

**Writing – review & editing:** Behrouz Ghazi Esfahani, Madelynn K. Bowman, Rebecca W. Alexander, M. Elizabeth Stroupe.

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
