## [Decision Letter · Decision Letter 0]

26 Dec 2025

PONE-D-25-61422Mycoplasma penetrans methionyl tRNA synthetase is an asymmetric dimer fused to N-terminal ancillary domainsPLOS One

Dear Dr. Stroupe,

Thank you for submitting your manuscript to PLOS ONE. After careful consideration, we feel that it has merit but does not fully meet PLOS ONE’s publication criteria as it currently stands. Therefore, we invite you to submit a revised version of the manuscript that addresses the points raised during the review process.

We look forward to receiving your revised manuscript.

Kind regards,

Shailender Kumar Verma, Ph.D.

Academic Editor

PLOS One

Journal Requirements:

“We thank Dr. Christopher Stroupe and Jen Kennedy for helpful discussions. Florida State University supports cryo-EM in the Biological Imaging Resource Center, which houses the following equipment used in this study: a Gatan Solaris Plasma Cleaner (NIH grant S10 RR024564), a Hitachi HT7800 (NSF grant MRI2017869 to M.E.S.), a ThermoFisher Vitrobot Mark 14 IV (NIH grant S10 RR024564), an SPI chameleon® plunging system (NIH grant R24 GM145964),a ThermoFisher Titan Krios (NIH grant S10 RR025080), and a DE Apollo direct electron detector (NIH grant R35 GM139616). This work was further supported by National Science Foundation grants MCB1856502 and CHE1904612 to M.E.S. “

“We thank Dr. Christopher Stroupe and Jen Kennedy for helpful discussions. Florida State University supports cryo-EM in the Biological Imaging Resource Center, which houses the following equipment used in this study: a Gatan Solaris Plasma Cleaner (NIH grant S10 RR024564), a Hitachi HT7800 (NSF grant MRI2017869 to M.E.S.), a ThermoFisher Vitrobot Mark 14 IV (NIH grant S10 RR024564), an SPI chameleon® plunging system (NIH grant R24 GM145964),a ThermoFisher Titan Krios (NIH grant S10 RR025080), and a DE Apollo direct electron detector (NIH grant R35 GM139616). This work was further supported by National Science Foundation grants MCB1856502 and CHE1904612 to M.E.S. “

“The author(s) received no specific funding for this work”

Reviewers' comments:

Reviewer's Responses to Questions

**Comments to the Author**

1. Is the manuscript technically sound, and do the data support the conclusions?

Reviewer #1: Partly

Reviewer #2: Yes

Reviewer #3: Partly

Reviewer #4: Partly

2. Has the statistical analysis been performed appropriately and rigorously?

Reviewer #1: I Don't Know

Reviewer #2: Yes

Reviewer #3: Yes

Reviewer #4: N/A

3. Have the authors made all data underlying the findings in their manuscript fully available?

Reviewer #1: Yes

Reviewer #2: Yes

Reviewer #3: Yes

Reviewer #4: Yes

4. Is the manuscript presented in an intelligible fashion and written in standard English?

Reviewer #1: Yes

Reviewer #2: Yes

Reviewer #3: Yes

Reviewer #4: Yes

5. Review Comments to the Author

Reviewer #1: Major revision

1. In the background, authors can state more about the progress of MetRS structure, and then introduce the aim of this study.

2. In fact, the Cryo-EM sample preparation is the main part of this study, which determined the final results, and it can describe the protocol in details.

3. It should compare current results with others, and show the unique of MetRS in structure and explain the potential impacts of Mycoplasma penetrans.

Minor revision

1. The resolution of figure 1 can be improved.

2. Figure 2 and 3 can integrate into one.

Reviewer #2: Manuscript ID: PONE-D-25-61422

Recommendation: Major Revision

Reviewer: Morufu Olalekan Raimi, PhD

Affiliation: Niger Delta Institute for Emerging and Re-emerging Infectious Diseases, Federal University, Otuoke, Bayelsa State

Review Date: 4-December, 2024

Dear Editors and Authors,

Thank you for the opportunity to review the manuscript titled “Mycoplasma penetrans methionyl tRNA synthetase is an asymmetric dimer fused to N-terminal ancillary domains.” The study presents a structural investigation of a unique chimeric MetRS enzyme from M. penetrans, employing cryo-EM to elucidate its domain architecture and conformational dynamics. The work addresses an interesting evolutionary and functional question in the context of streamlined genomes and domain fusion events. While the structural data and overall premise are promising, several significant concerns must be addressed before the manuscript can be considered for publication.

i. Title and Abstract

• Title: The title is clear and accurately reflects the content.

• Abstract: The abstract succinctly summarizes the study but could be strengthened by explicitly stating the resolution of the cryo-EM map and the functional implications of the observed asymmetry. Currently, the abstract reads as descriptive; it should better highlight the novelty e.g., the first structure of a triple-domain MetRS fusion and its implications for substrate channeling in a minimal genome organism.

ii. Introduction

The introduction adequately contextualizes M. penetrans and the peculiarity of its metS gene. However, it lacks a clear thematic bridge between domain fusion, structural asymmetry, and functional adaptation in parasitic bacteria.

• Author should expand on why structural asymmetry might be evolutionarily advantageous in this context. Additionally, briefly introduce the concept of substrate channeling in fused enzymes to frame the study’s hypothesis more explicitly.

iii. Materials and Methods

This part is generally well-described but requires clarification in several areas:

1. Sample Preparation:

o The concentration of MpMetRS used for cryo-EM (0.04 mg/mL) is extremely low for typical cryo-EM studies. Justify this choice and discuss potential implications for particle distribution and ice quality.

o Author should clarify whether the “5-fold excess folded tRNA” refers to molar excess and whether tRNA was pre-incubated under aminoacylation conditions.

2. Data Processing:

o The transition from symmetric (C2) to asymmetric refinement due to “ill-defined additional features” is not sufficiently justified. Provide quantitative metrics (e.g., FSC curves, particle distribution) to support this decision.

o The absence of density for the AARS catalytic domain is a major limitation. Discuss whether this is due to flexibility, partial occupancy, or resolution limits. Include a supplemental figure showing local resolution maps.

3. Model Validation:

o Provide detailed validation statistics (CC, Ramachandran outliers, clash scores) for the deposited model (9QS7).

o The use of AlphaFold2 to model a symmetric dimer is interesting but should be framed as speculative. Clearly distinguish between experimental observations and computational predictions.

iv. Results

1. Structural Findings:

o The description of the AGAT dimer interface is clear, but the analysis of conserved vs. unique interactions would benefit from a quantitative comparison (e.g., buried surface area, interaction energies).

o The claim that the N-terminal domain is a nucleotidyl transferase is based on DALI results but lacks experimental validation (e.g., activity assays). This should be toned down or explicitly noted as a prediction.

2. Asymmetry and Mobility:

o The observation that only the N-terminal and AGAT domains follow C2 symmetry is intriguing. However, the manuscript does not sufficiently explore the functional implications of this asymmetry. Is the asymmetry fixed or dynamic?

o The missing density for one AARS domain is a critical point. Consider whether this could be due to proteolytic cleavage or flexible linkage. SDS-PAGE (Fig S2) should be referenced here to confirm full-length protein integrity.

3. Oligomerization at High Concentration:

o The reported tetramer/hexamer formation is interesting but preliminary. Provide size-exclusion chromatography or analytical ultracentrifugation data to corroborate the oligomeric state in solution.

4. Figures:

o Figures are generally informative but could be improved:

Fig 3: Label the domains consistently (NTase, AGAT, AARS).

Fig 6: The docking models are useful but should include a measure of uncertainty (e.g., RMSD from multiple docking runs).

v. Discussion

The discussion is somewhat speculative and should be more tightly linked to the structural observations.

1. Functional Interpretation:

o The proposal that the NTase domain “increases local ATP concentration” is interesting but untested. Suggest future experiments (e.g., ATPase activity assays) or temper the claim.

o The idea of substrate channeling between AGAT and MetRS domains is appealing but not supported by direct evidence. A more cautious interpretation is warranted.

2. Evolutionary Context:

o Expand on why such a complex fusion evolved in a genome-reduced organism. Could this be a compensatory mechanism for host dependence?

3. Limitations:

o The resolution (3.66 Å) is moderate for detailed mechanistic insights, especially for flexible regions. Acknowledge this limitation.

o The lack of a tRNA-bound high-resolution structure limits functional conclusion. Mention this as a key area for future work.

vi. Data Availability

The deposition of coordinates (9QS7) and maps (EMD-70794) is commendable. Ensure that the half-maps and processing metadata are also publicly accessible to allow independent validation.

vii. Overall Assessment

This study presents a novel and interesting structure of a chimeric MetRS with potential implications for understanding domain fusion and enzyme adaptation in minimal genomes. The cryo-EM work is technically sound, but the manuscript currently overinterprets the data in several places and lacks critical validations.

Major revisions are required to:

1. Clarify methodological details, especially regarding sample preparation and data processing choices.

2. Temper functional claims that are not experimentally supported.

3. Strengthen the discussion by linking structural observations more directly to possible biological mechanisms.

4. Improve the presentation of figures and models to better convey asymmetry and conformational flexibility.

With these revisions, the manuscript could make a valuable contribution to the fields of structural biology and molecular evolution.

Recommendation: Major Revision

Sincerely,

Dr. Morufu Olalekan RAIMI,

BSc, (Geography. and Environmental Management), Diploma. (Environmental Health), M.Sc. Environmental Health Management), M.Phil. (Environmental Health Science), P.hD (Environmental Health Science), MNES, REHO, LEHO, FAIWMES

Environmental Health Consultant/Lecturer at Federal University Otuoke, Bayelsa State. Nigeria.

Environmental Health Consultant to United Nations Economic Commission for Europe (UNECE) Expert Group on Resources Management (EGRM). Geneva, Switzerland.

Research Consultant to Bayelsa State Primary Health Care Board.

Former Technical Adviser to the Executive Secretary, Bayelsa State Primary Health Care Board.

Former Director, Advocacy, Communication and Social Mobilization, Bayelsa State Primary Health Care Board.

Program Manager, Centre for Niger Delta Studies and Sustainability (CNDSS), Federal University Otuoke, Bayelsa State.

Deputy Director, Niger Delta Institute for Emerging and Re-emerging Infectious Diseases (NDIERID), Federal University Otuoke, Bayelsa State.

Reviewer to National Science Foundation (NSF) Graduate Research Fellowship Program (GRFP)

Plos One Academic Editor

https://publons.com/a/1479339/

https://ssrn.com/author=2891311

ORCID iD: https://orcid.org/0000-0001-5042-6729

Web of Science Researcher ID: https://publons.com/a/1479339/

Website: https://ssrn.com/author=2891311; https://www.growkudos.com/profile/morufu_raimi; https://sciprofiles.com/profile/Morufuolalekanraimi; https://livedna.org/234.27529

https://scholar.google.com/citations?user=nRBW82AAAAAJ&hl=en.

https://theconversation.com/profiles/morufu-olalekan-raimi-1520774

Reviewer #3: The manuscript by B.G. Esfahani and co-authors investigates the structure of the methionyl-tRNA synthetase (MetRS) from the pathogen Mycoplasma penetrans (Mp). Despite the highly reduced genome of this bacterium, MpMetRS is an unusually large protein (1087 amino acids, 126.4 kDa), owing to an additional module appended to the N-terminus of the canonical MetRS enzyme. Previous biochemical studies have shown that full-length MpMetRS is expressed in vivo, with its N-terminal module mediating dimerization in vitro and containing an alanine-glyoxylate aminotransferase (AGAT) domain involved in methionine biosynthesis. Notably, the methionine produced by the AGAT domain can be used by the MetRS domain to synthesize Met-tRNAMet in vitro, suggesting potential substrate channeling and an intriguing evolutionary adaptation.

In the present study, authors determined the structure of MpMetRS at 3.66Å resolution using single-particle cryo-electron microscopy (cryo-EM), revealing a three-domain architecture comprising an N-terminal nucleotidyl transferase (NTase) domain (residues 1 to 166), followed by the AGAT domain (residues 167 to 560), and the C-terminal MetRS domain (residues 561 to 1087). The identification of the N-terminal NTase domain represents a major contribution of this work, providing a more complete view of the modular organization of MpMetRS. The cryo-EM structure is consistent with the previously reported dimeric nature of MpMetRS and confirms that dimerization is driven by the AGAT domain, whose mode of association closely resembles that observed in related AGAT proteins (e.g. Anabaena), while also displaying interactions specific to M. penetrans. The NTase and AGAT domain adopt a well-defined two-fold symmetric arrangement, whereas the MetRS domain is only partially resolved and appears asymmetrically positioned. Despite extensive attempts to visualize tRNA-bound states, no well-defined density corresponding to bound tRNA or to fully resolved MetRS domain could be obtained, indicating significant conformational flexibility of this region. Based on docking of orthologous MetRS-tRNA complexes and AlphaFold2 modeling, the authors propose that significant domain arrangements would be required to accommodate tRNA. Finally, at higher protein concentration and using a different strategy for grid preparation, the authors observed higher-order oligomeric assemblies involving the NTase and AGAT domains, although their biological relevance remains unclear. Overall, the study provides structural insights into the organization, symmetry, and oligomeric state of this multidomain MetRS enzyme, while emphasizing the dynamic nature of its catalytic domain.

Although this work is of undeniable interest, some concerns must be addressed to improve the presentation and interpretation of results.

Major concerns:

1) The main strength of this study is the resolution of the structure of the N-terminal module of MpMetRS, which consists of an NTase domain followed by an AGAT domain. The identification of the NTase domain is a particularly important contribution, as this domain had not been annotated on the modular description of the enzyme. However, a description of the overall architecture of the NTase domain is missing from the Results section and is only briefly mentioned in the Discussion section. Similarly, although the dimerization interface of the MpMetRS AGAT domain is described and compared with Anabaena AGAT dimer, a description of the overall MpMetRS AGAT structure and the “canonical” dimerization mode of this class of enzymes would improve the clarity and completeness of this section.

2) The successful application of single-particle cryo-EM to a large, flexible, and multidomain enzyme such as MpMetRS is challenging. While the NTase and AGAT domains were fully resolved, only the C-terminal anticodon-binding domain of one of the two MpMetRS protomers could be modeled based on experimental data. The N-terminal module of MpMetRS, comprising the NTase and AGAT domains, dimerizes through the AGAT domain, resulting in a stable dimeric core with C2 symmetry. However, it is unclear whether the asymmetrical position of the observed C-terminal MetRS domain reflects true structural asymmetry of the MpMetRS enzyme or simply flexibility. Additional discussion or evidence is needed to clarify this point. The title of the study might be reconsidered.

3) The observation of higher-order oligomeric assemblies at elevated protein concentration and under specific grid preparation conditions is fascinating, but their biological relevance remains to be investigated. It is unclear whether these oligomers exist in solution under native conditions or arise as an artifact of cryo-EM grid preparation, and the authors should clarify this point. In the absence of additional experiments (e.g. size-exclusion chromatography, dynamic light scattering) supporting concentration-dependent oligomerization of NTase-AGAT units, the potential functional role of these oligomers should be more clearly emphasized as speculative. I suggest presenting these observations after the “Modeling” section in Results and condensate Figures 5 and 7 into one.

4) Despite extensive efforts, no clear density was observed for the MetRS domain in complex with tRNA. Nevertheless, the authors discuss steric constraints and domain rearrangements required for tRNA binding based on docking with orthologous MetRS-tRNA complexes and on AlphaFold2 predictions. While these approaches are reasonable, they cannot substitute for experimental validation. Techniques such as small-angle X-ray scattering (SAXS) could provide evidence for the large conformational changes associated with tRNA binding and would strengthen the study. In the absence of such additional data, authors should more explicitly acknowledge the limitations of the study and further emphasize the speculative nature of the proposed mechanisms for tRNA binding. In addition, I suggest including and discussing AlphaFold3 models of MpMetRS alone and in complex with tRNAMet, which might provide further insights into these potential structural rearrangements.

5) The authors propose potential functional coupling between NTase, AGAT, and MetRS domains as an evolutionary advantage of this multidomain MetRS enzyme. While this hypothesis is interesting and appropriately presented as speculative, the concluding paragraph does not sufficiently emphasize the main contribution of the study that is directly supported by experimental data: the structure of the M. penetrans-specific N-terminal module of MetRS, comprising an NTase and AGAT domain that drives dimerization. The authors should refocus the conclusion and integrate the hypothesis of functional coupling withing the Discussion section.

Minor concerns:

1) A sentence summarizing the results of the study by Jones. T.E. et al. (2008) (doi : 10.1016/j.molcel.2007.12.021) should be included in the introduction. It would be useful to state that the N-terminal module of MpMetRS is not involved in the unusual mechanism of tRNA substrate discrimination described in that study.

2) The sentence at line 77 of the Introduction appears redundant, as the preceding sentence already states that the N-terminal-most domain corresponds to an NTase domain.

3) In the sentence at line 78 of the Introduction, the authors should explicitly indicate which enzymes domains are organized around the AGAT domain.

4) In the sentence at line 84 of the Material and Methods section, the authors should add the GeneBank identifier of the metS gene and justify the use of the M568A MpMetRS mutant for the structural analysis.

5) Showing the elution profile of MpMetRS on the Sephacryl S300 size-exclusion column as a supplementary figure might provide additional evidence for conformational heterogeneity.

6) Including additional AGAT sequences, modeled by AlphaFold2 or AlphaFold3, in the structure-based alignment shown in Figure S1 could help better identify interactions that are conserved among AGAT dimers and those that are specific to Mycoplasma.

7) In the legend of Figure 1, the inset is not described.

8) In the legend of Figure 3, the MetRS domain displayed in cartoon representation is not mentioned. The name of the different ligands shown in spheres should be also mentioned.

9) A scheme representing the modular organization of MpMetRS would strengthen the clarity of the manuscript. This might be included as part of Figure 1.

10) Several figures in the Results section are difficult to interpret (e.g. Figures 1, 4, and 5). Their clarity could be improved by using color coding for the different domains, indicating symmetry axes, and including schematic diagrams that illustrate the overall domain organization (as shown in Figure 5B).

11) Figure 7 and its legend need to be improved by showing also the cartoon representation of the concatenated NTase-AGAT subunits. The nature of the ligands in the active sites should be stated in the legend. The region zoomed should be described.

12) Provide supplementary tables summarizing cryo-electron microscopy data collection parameters and refinement statistics.

Reviewer #4: The contribution entitled Mycoplasma penetrans methionyl tRNA synthetase is an asymmetric dimer fused to N-terminal ancillary domains by Esfahani et al., reports a partial structure of the unusual Mycoplasma penetrans methionyl-tRNA synthetase (MpMetRS) at a 3.7 A resolution. The MpMetRS has been previously shown to contain an aminotransferase domain fusion which is able to catalyze conversion of 2-keto-4-methylthiobutyrate (KMTB) to methionine and that this methionine can be subsequently transferred to tRNAMet in vitro (Muraski et al., 2020). Here, the authors were able to solve the structure of the aminotransferase domain along with the N-terminal nucleotidyl transferase domain. The topic is highly interesting, particularly the functional roles of fusion enzymes in minimal-genome organisms and the enrichment of enzymatic capabilities through the acquisition of novel domains. However, the manuscript is difficult to follow, as key background information is missing and the conclusions are not always fully supported by the presented results.

Major comments:

Abstract: From the abstract, it is not clear which findings are directly supported by the data and which are speculative. In addition, the simultaneous use of the full-length MetRS and the isolated MetRS domain is somewhat confusing.

Introduction: The background information on MetRS is in my opinion insufficient. For example, the manuscript does not clearly describe the canonical architecture of MetRS enzymes, or identify which known MetRS enzyme is the MpMetRS most closely related to. It is unclear whether it is expected to function as a monomer or dimer. It is also unclear whether the activity of the N-terminal NTase domain has been previously established. A schematic domain diagram of MpMetRS would greatly aid clarity, as the current text makes it difficult to distinguish actual architecture of the MpMetRS (e.g., it is sometimes difficult to understand whether it is the N-terminal NTase domain or the N-terminal region of the MetRS (“MetRS domain”)). In addition, the final sentence of the Introduction is not very clear and would benefit from further elaboration.

Results: Overall, it remains difficult to discern what is definitively demonstrated in the manuscript. For example, the assignment of the “additional density with helical features” to the C-terminal anticodon-binding domain is not sufficiently justified, and the level of confidence in this interpretation is unclear. Similarly, the rationale for modeling the tRNA using E. coli CysRS:tRNACys rather than the MetRS:tRNAMet structure (Nakanishi et al., 2005) is not explained.

Several figure legends are incomplete, and Figure 6 in particular is difficult to interpret, as the aligned and clashing regions are not clearly presented. The conclusion that MpMetRS forms a dimer (line 210) is also not well supported, given that the native gel (Fig. S2) suggests species closer to trimers or tetramers and that tetrameric assemblies are observed in cryo-EM at higher concentrations.

In addition, it is unclear why tRNA-containing samples were not examined using the chameleon plunging system, or whether attempts were made to isolate a defined MpMetRS:tRNAMet complex prior to cryo-EM, rather than relying on excess tRNA during grid preparation.

6. PLOS authors have the option to publish the peer review history of their article (what does this mean?). If published, this will include your full peer review and any attached files.

Reviewer #1: No

Reviewer #2: **Yes:**Morufu Olalekan Raimi Ph.D

Reviewer #3: No

Reviewer #4: No

---

## [Author Response · Author response to Decision Letter 1]

25 Feb 2026

Florida State University • Tallahassee • FL 32310

Biological Science • Inst. Mol. Biophysics

Telephone: (850) 644-1751 FAX: (850) 644-1366 e-mail: mestroupe@bio.fsu.edu

February 16, 2026

Dr. Shailender Kumar Verma, Editor

PlosOne

Dear Dr. Verma,

We are pleased to resubmit our newly titled manuscript, “Mycoplasma penetrans methionyl-tRNA synthetase dimerizes via tandem N-terminal ancillary domains”. We used cryogenic electron microscopy to determine the structure of an unusual methionyl tRNA synthetase (MetRS) from the opportunistic pathogen M. penetrans.

We feel we have responded to each of the reviewers’ responses to result in an improved manuscript, as outlined in the response-to-reviewers.

The funders had no role in these studies: "The funders had no role in study design, data collection and analysis, decision to publish, or preparation of the manuscript."

Please see below our response to reviewers.

Thank you for your support of our contribution,

M. Elizabeth Stroupe, PhD

Professor, Department of Biological Science and Institute of Molecular Biophysics

Florida State University

Rebecca W. Alexander, PhD

Associate Dean for Research and Community Engagement

Professor, Department of Chemistry

Wake Forest University

Journal Requirements:

The title page has been formatted according to the style requirements.

No unique code was used in this study.

“We thank Dr. Christopher Stroupe and Jen Kennedy for helpful discussions. Florida State University supports cryo-EM in the Biological Imaging Resource Center, which houses the following equipment used in this study: a Gatan Solaris Plasma Cleaner (NIH grant S10 RR024564), a Hitachi HT7800 (NSF grant MRI2017869 to M.E.S.), a ThermoFisher Vitrobot Mark 14 IV (NIH grant S10 RR024564), an SPI chameleon® plunging system (NIH grant R24 GM145964),a ThermoFisher Titan Krios (NIH grant S10 RR025080), and a DE Apollo direct electron detector (NIH grant R35 GM139616). This work was further supported by National Science Foundation grants MCB1856502 and CHE1904612 to M.E.S. “

The statement is correct, and we have included it in our revised cover letter.

“We thank Dr. Christopher Stroupe and Jen Kennedy for helpful discussions. Florida State University supports cryo-EM in the Biological Imaging Resource Center, which houses the following equipment used in this study: a Gatan Solaris Plasma Cleaner (NIH grant S10 RR024564), a Hitachi HT7800 (NSF grant MRI2017869 to M.E.S.), a ThermoFisher Vitrobot Mark 14 IV (NIH grant S10 RR024564), an SPI chameleon® plunging system (NIH grant R24 GM145964),a ThermoFisher Titan Krios (NIH grant S10 RR025080), and a DE Apollo direct electron detector (NIH grant R35 GM139616). This work was further supported by National Science Foundation grants MCB1856502 and CHE1904612 to M.E.S. “

“The author(s) received no specific funding for this work”

We have updated the funding statement, as presented in the cover letter.

We have moved this information to the Funding Information section.

Noted. Thank you

Reviewers' comments:

Reviewer's Responses to Questions

Comments to the Author

1. Is the manuscript technically sound, and do the data support the conclusions?

Reviewer #1: Partly

Reviewer #2: Yes

Reviewer #3: Partly

Reviewer #4: Partly

2. Has the statistical analysis been performed appropriately and rigorously?

Reviewer #1: I Don't Know

Reviewer #2: Yes

Reviewer #3: Yes

Reviewer #4: N/A

3. Have the authors made all data underlying the findings in their manuscript fully available?

Reviewer #1: Yes

Reviewer #2: Yes

Reviewer #3: Yes

Reviewer #4: Yes

4. Is the manuscript presented in an intelligible fashion and written in standard English?

Reviewer #1: Yes

Reviewer #2: Yes

Reviewer #3: Yes

Reviewer #4: Yes

5. Review Comments to the Author

Reviewer #1: Major revision

1. In the background, authors can state more about the progress of MetRS structure, and then introduce the aim of this study.

We have included this information in new paragraph 5, Introduction:

“AARSs from diverse taxa have been biochemically dissected to explain their ancient and essential role in faithfully translating the genetic code, so much is known about the induced fit mechanism that allows the enzyme to discriminate amongst similar amino acid functional groups to pair them with a three-nucleotide anticodon. Nevertheless, high-resolution structure determination techniques rely on specimen homogeneity, so many of the existing AARS structures have missing features that do not fully define the structural basis for the AARS mechanism. For example, tRNAs that are aminoacylated by class I sub-family AARSs necessarily undergo a “hairpin” rearrangement of the 3’-terminal acceptor end to approach the activated amino acid in the AARS catalytic site. Few high-resolution structures of such hairpinned tRNAs bound to their cognate AARSs are available. The modular nature of AARSs contributes to efficient catalysis and a wide array of novel functions but a complete model of the structural basis for the biochemical data remains elusive. In the case of MpMetRS, ambiguity in sequence homology of the N-terminal domain of unknown function necessitated further experimental analysis.”

2. In fact, the Cryo-EM sample preparation is the main part of this study, which determined the final results, and it can describe the protocol in details.

We have added the following details to the Materials and Methods:

“Further refinement of the apo-MpMetRS dataset by template-free 2D classification identified 202,593 particles with clear 2-fold symmetry that were used for initial model building. Non-uniform 3D refinement with imposed C2 symmetry resulted in a 3.27 Å-resolution map. The map showed no evidence of the MetRS catalytic domain. Closer inspection of the 2D classes revealed some classes with blurred features that did not follow the 2-fold symmetry that dominated the initial classification. To explore the possibility that this density corresponded to the missing MetRS catalytic domain, this limited subset of particles (60,000) was used to refine a non-uniform, asymmetric map to 3.66 Å resolution. All maps were sharpened with deepEMhancer for enhanced visualization .”

3. It should compare current results with others, and show the unique of MetRS in structure and explain the potential impacts of Mycoplasma penetrans.

We have expanded our analysis to compare in detail the N-terminal domains (the domain of unknown function) with homology to other structural homologs.

Minor revision

1. The resolution of figure 1 can be improved.

We have regenerated all figures as tif files to a resolution of 300 d.p.i. with a width of 5.2 inches.

2. Figure 2 and 3 can integrate into one.

We have reorganized the figures to enhance the readability and flow of the manuscript.

Reviewer #2: Manuscript ID: PONE-D-25-61422

Recommendation: Major Revision

Reviewer: Morufu Olalekan Raimi, PhD

Affiliation: Niger Delta Institute for Emerging and Re-emerging Infectious Diseases, Federal University, Otuoke, Bayelsa State

Review Date: 4-December, 2024

Dear Editors and Authors,

Thank you for the opportunity to review the manuscript titled “Mycoplasma penetrans methionyl tRNA synthetase is an asymmetric dimer fused to N-terminal ancillary domains.” The study presents a structural investigation of a unique chimeric MetRS enzyme from M. penetrans, employing cryo-EM to elucidate its domain architecture and conformational dynamics. The work addresses an interesting evolutionary and functional question in the context of streamlined genomes and domain fusion events. While the structural data and overall premise are promising, several significant concerns must be addressed before the manuscript can be considered for publication.

i. Title and Abstract

• Title: The title is clear and accurately reflects the content.

• Abstract: The abstract succinctly summarizes the study but could be strengthened by explicitly stating the resolution of the cryo-EM map and the functional implications of the observed asymmetry. Currently, the abstract reads as descriptive; it should better highlight the novelty e.g., the first structure of a triple-domain MetRS fusion and its implications for substrate channeling in a minimal genome organism.

We added the following to the abstract:

“We used cryo-EM to analyze the structure of the MpMetRS gene product to show that it is the product of three distinct enzyme domains: an N-terminal domain (NTD) of unknown function, a dimeric alanine-glyoxylate aminotransferase, and a MetRS. Only the first two N-terminal domains show two-fold symmetry, which were resolved to 3.27 Å resolution and the MetRS domain is only partially resolved to 3.66 Å resolution. Modelling the full structure shows that a conformational change must occur to accommodate a tRNA-bound MetRS domain. Further rearrangement of the catalytic domains would also be necessary to bring the active sites adjacent to one another if this unique assembly of catalytic domains functions to channel substrates to MetRS.”

ii. Introduction

The introduction adequately contextualizes M. penetrans and the peculiarity of its metS gene. However, it lacks a clear thematic bridge between domain fusion, structural asymmetry, and functional adaptation in parasitic bacteria.

• Author should expand on why structural asymmetry might be evolutionarily advantageous in this context.

Additionally, briefly introduce the concept of substrate channeling in fused enzymes to frame the study’s hypothesis more explicitly.

We thank the reviewer for these helpful suggestions. As we do not have direct observations of a substrate channeling complex, we opted to enhance our discussion of the variety of AARS fusions in the literature as follows to new paragraph 5, Introduction:

“AARSs from diverse taxa have been biochemically dissected to explain their ancient and essential role in faithfully translating the genetic code. Many AARSs exhibit induced fit, both to select their cognate tRNAs based on nucleotide identities in the three-nucleotide anticodon and the acceptor stem and to discriminate amongst similar amino acid functional groups. Nevertheless, high-resolution structure determination techniques rely on specimen homogeneity, so many of the existing AARS structures have missing features that do not fully define the structural basis for the AARS mechanism. For example, tRNAs that are aminoacylated by class I sub-family AARSs necessarily undergo a “hairpin” rearrangement of the 3’-terminal acceptor end to approach the activated amino acid in the AARS catalytic site. Few high-resolution structures of such hairpinned tRNAs bound to their cognate AARSs are available. The modular nature of AARSs contributes to efficient catalysis and a wide array of novel functions but a complete model of the structural basis for the biochemical data remains elusive. In the case of MpMetRS, ambiguity in sequence homology of the N-terminal domain of unknown function necessitated further experimental analysis.”

iii. Materials and Methods

This part is generally well-described but requires clarification in several areas:

1. Sample Preparation:

o The concentration of MpMetRS used for cryo-EM (0.04 mg/mL) is extremely low for typical cryo-EM studies. Justify this choice and discuss potential implications for particle distribution and ice quality.

We extensively screened protein concentrations and found that the protein aggregates (or polymerizes, see below) at high concentrations. The graphene substrate adheres to the protein, thus concentrating it in the ice before blotting and plunge freezing. To clarify, we added the following to Materials and Methods/Cryo-EM sample preparation:

“MpMetRS adheres to the graphene surface, allowing low concentration to avoid protein aggregation.”

o Author sh

---

## [Decision Letter · Decision Letter 1]

22 Mar 2026

PONE-D-25-61422R1Mycoplasma penetrans methionyl-tRNA synthetase dimerizes via tandem N-terminal ancillary domainsPLOS One

Dear Dr. Stroupe,

Thank you for submitting your manuscript to PLOS ONE. After careful consideration, we feel that it has merit but does not fully meet PLOS ONE’s publication criteria as it currently stands. Therefore, we invite you to submit a revised version of the manuscript that addresses the points raised during the review process.

We look forward to receiving your revised manuscript.

Kind regards,

Shailender Kumar Verma, Ph.D.

Academic Editor

PLOS One

Journal Requirements:

Reviewers' comments:

Reviewer's Responses to Questions

**Comments to the Author**

1. If the authors have adequately addressed your comments raised in a previous round of review and you feel that this manuscript is now acceptable for publication, you may indicate that here to bypass the “Comments to the Author” section, enter your conflict of interest statement in the “Confidential to Editor” section, and submit your "Accept" recommendation.

Reviewer #2: All comments have been addressed

Reviewer #3: All comments have been addressed

Reviewer #4: All comments have been addressed

2. Is the manuscript technically sound, and do the data support the conclusions?

Reviewer #2: Partly

Reviewer #3: Yes

Reviewer #4: Yes

3. Has the statistical analysis been performed appropriately and rigorously?

Reviewer #2: Yes

Reviewer #3: Yes

Reviewer #4: Yes

4. Have the authors made all data underlying the findings in their manuscript fully available?

Reviewer #2: Yes

Reviewer #3: Yes

Reviewer #4: Yes

5. Is the manuscript presented in an intelligible fashion and written in standard English?

Reviewer #2: Yes

Reviewer #3: Yes

Reviewer #4: Yes

6. Review Comments to the Author

Reviewer #2: Manuscript Number: PONE-D-25-61422R1

Title: Mycoplasma penetrans methionyl-tRNA synthetase dimerizes via tandem N-terminal ancillary domains

Recommendation: Minor Revision

Review Date: 2026-03-13

Reviewer Comments

Dear Authors,

I. Title

The revised title accurately reflects the study's main finding.

• Current Title: “Mycoplasma penetrans methionyl-tRNA synthetase dimerizes via tandem N-terminal ancillary domains”

The title is clear, descriptive, and appropriately focused on the key structural insight—that dimerization is mediated by the N-terminal domains (NTD and AGAT). This is a significant improvement over the previous title, which emphasized asymmetry. The current title is more accurate given that asymmetry is inferred from partial density rather than directly observed for both domains.

• Recommendation: Retain the current title.

II. Abstract

The revised abstract is much improved and now clearly distinguishes experimental findings from modeling.

• Minor Issues:

1. Phrasing: “We used cryo-EM to analyze the structure of the metS gene product (MpMetRS) to show that it is the product of three distinct enzyme domains” – slightly redundant (“product of the gene product”). Consider: “We used cryo-EM to determine the structure of the MpMetRS protein, revealing that it comprises three distinct enzyme domains.”

2. Specificity: The abstract mentions “Modelling the full structure shows that a conformational change must occur.” It might be helpful to briefly indicate what kind of change (e.g., “rotation of the MetRS domain relative to the AGAT core”).

• Recommendation: Minor refinements as suggested.

III. Introduction

The introduction is now comprehensive and well-structured, with clear progression from biological context to specific aims.

• Minor Issues:

1. Citation formatting: Some references in the introduction (e.g., references 3, 8) are cited with “Muraski et al., 2020” and “Jones et al., 2008” – these appear correctly in the reference list, but ensure all are consistently formatted.

2. Terminology consistency: The introduction uses both “NTD” and “NTase” for the N-terminal domain. Table 1 clarifies that NTD is the domain name and NTase is the predicted function. This is helpful, but the introduction could briefly note that “based on structural homology, we identify this domain as a nucleotidyl transferase (NTase) fold, though its specific substrate remains unknown.”

3. Final paragraph: The final sentence (“This is the first example, to our knowledge, of this collection of domains in a MetRS, allowing speculation that the metabolic functions may be coupled...”) is appropriate. However, it could be slightly more specific: “allowing speculation that the three activities, nucleotide metabolism, methionine biosynthesis, and tRNA aminoacylation, may be functionally coupled in this minimal-genome organism.”

• Recommendation: Minor refinements as suggested.

IV. Materials and Methods

The methods section is thorough and well-organized. The additions in response to reviewers have addressed previous concerns.

• Minor Issues:

1. tRNA preparation: The description of tRNA preparation is adequate but could be slightly expanded: "tRNA was resuspended in TE buffer; concentration was determined by absorbance at 260 nm." It might be helpful to note the extinction coefficient used or the approximate yield.

2. Cryo-EM sample preparation: The description of the two preparation methods (low concentration on graphene grids vs. high concentration with chameleon) is clear. However, the sentence "MpMetRS adheres to the graphene surface, allowing low concentration to avoid protein aggregation” could be moved earlier for clarity.

3. Data processing: The description of the transition from symmetric to asymmetric refinement is now well-justified with reference to 2D classes and GSFSC curves (S1 and S2 Figs). Good.

4. Model building and refinement: The description of model building with ModelAngelo and refinement in PHENIX is appropriate. The DALI searches are well-described. However, it would be helpful to include a brief statement about model validation statistics (e.g., “Refinement statistics, including map-model correlations, Ramachandran statistics, and clash scores, are provided in Supplemental Table 1”).

5. Dynamic light scattering: The description is clear. Minor: “Data was analyzed” should be “Data were analyzed” (plural).

• Recommendation: Minor refinements as suggested, particularly adding model validation statistics to the methods or referring to them in Supplemental Table 1.

V. Results

The results are clearly presented and well-organized. The new figures and tables enhance readability.

• 5.1. The NTD has structural homology to an NTase: This section is excellent. The DALI results (Table 2) convincingly demonstrate the structural relationship. Figure 2 is clear and well-annotated. The description of the conserved core and divergent loops is precise and informative. The observation about the closed loop potentially occluding the nucleotide-binding pocket is interesting and appropriately noted.

• Minor: The sentence “Regions that are non-conserved are gray” in the Figure 2 legend is clear. In the text, the authors might briefly speculate whether the closed loop conformation is functionally relevant or an artifact of the apo state.

• 5.2. The central domain is a dimeric AGAT: This section is similarly strong. Table 3 provides a useful overview of structural homologs. Figure 3 is clear, and the description of the dimer interface (with buried surface area and specific interactions) is appropriately quantitative. The comparison with TdPntC-AEPT (PDBID 6PD1) is illuminating and well-illustrated in Figure 3C.

• Minor: The sentence “In each homolog, the dimer interface buries about 3504 Å² (of a total of 23,060 Å² for each monomer)” – the total surface area for each monomer seems high (23,060 Å²). Is this the correct value? A typical 60 kDa domain might have ~20,000 Å² total surface area, but the AGAT domain is ~44 kDa (residues 167-560). Please verify.

• 5.3. The position of the MetRS domain does not follow C2 symmetry: This section presents the key finding of asymmetry. The description is clear, and the authors appropriately note the limitations (limited resolution, no density for the second MetRS domain). The reference to PAGE analysis confirming full-length protein (S6A Fig) is important to rule out proteolysis. The observation of new 2D classes with tRNA (S7 Fig) is intriguing, even if high-resolution features were not obtained.

• Minor: The sentence “There is no well-resolved density for the N-terminal catalytic site of MetRS or for MetRS from the opposing subunit” – it might be helpful to explicitly state that this means the visible helical bundle corresponds to only one of the two MetRS domains, and that it is the C-terminal anticodon-binding domain, not the catalytic domain.

• 5.4. Modelling: This section appropriately uses docking and AlphaFold2 to explore conformational possibilities. The rationale for using EcCysRS (PDBID 1U0B) as a template because it captures the hairpinned tRNA acceptor stem is well-explained. The predicted steric clash and the proposed ~60° rotation are clearly described. The RMSD values provide appropriate uncertainty measures.

• Minor: The sentence “Docking the helical bundle from monomeric E. coli MetRS (PDBID 1QDT) [31] predicts that the catalytic domain would fall immediately between the C2-symmetric NTDs (Fig 5B) (R.M.S.D of 1.0 Å for a core of 108 residues).” – “R.M.S.D.” is typically written as “RMSD”. Also, ensure figure callouts are consistent (Fig 5B, 5C, 5D – the text refers to Fig 6 in places; please verify).

• 5.5. MpMetRS oligomeric state depends on the protein concentration: This section presents the intriguing observation of higher-order oligomers at high concentration. The SEC and DLS data (S6 Fig) provide valuable solution-state context. The comparison with TdPntC-AEPT in Figure 7 is appropriate. The authors appropriately note that the in vivo significance remains to be determined.

• Minor: The sentence “Within those higher-order oligomers, the NTD and AGAT domains appear concatenated, with the NTD domain of one polypeptide positioned in opposition to the AGAT domain of the next polypeptide to bury an additional 1575 Å².” – “NTD domain” is redundant (NTD already means N-terminal domain). Use “NTD” alone.

• Recommendation: Minor refinements as suggested, including verification of the surface area value and consistent figure referencing.

VI. Discussion

The discussion is now more focused and appropriately cautious. The revisions have addressed previous concerns about overinterpretation.

• Minor Issues:

1. Paragraph 2 (lines ~75-80 in the discussion): “To further explore the possibility that the active sites could be coupled to increase the local concentration of Met for aminoacylating tRNA, we modeled the apo anticodon binding subdomain based on the crystal structure of EcCysRS [31] (Fig 6).” – This should reference the correct figure (likely Fig 5 based on the revised numbering). Please verify all figure callouts.

2. Paragraph 3: “Prior molecular dynamic simulations performed on a monomeric form of EcMetRS... predicted that the zinc-binding insertion... are particularly mobile.” – This is a helpful reference. The authors might also note that the mobility of these regions explains why they are not visualized in the cryo-EM structure, which is an average.

3. Paragraph 4: “When the same docking is performed using a tRNA-bound AARS, EcCysRS [32], a conflict arises from the tRNA binding pose in relationship to the cleft between the central, C2-symmetric NTD/AGAT domain.” – The phrase "in relationship to" is awkward. Consider “relative to.”

4. Paragraph 5: “We further observed a concentration-dependent effect on the oligomeric state of MpMetRS.” – The discussion of oligomerization is appropriate, but the authors might briefly note whether similar concentration-dependent oligomerization has been observed for other NTase or AGAT family members.

5. Paragraph 6 (comparison with TdPntC-AEPT): This comparison is valuable. The authors might consider whether the different oligomerization modes (NTD-AGAT concatenation in MpMetRS vs. different interface in TdPntC-AEPT) might relate to the different domain organizations (NTD-AGAT-MetRS vs. PntC-AEPT).

6. Limitations: The discussion acknowledges the moderate resolution and lack of tRNA-bound structure. However, it could also explicitly note that the functional assignments (NTase activity) are based on structural homology and await biochemical validation. This is already implied but could be stated directly.

• Recommendation: Minor refinements as suggested, particularly ensuring consistent figure referencing and considering a brief note about the need for biochemical validation of the predicted NTase activity.

VII. Conclusion

The conclusion is concise and appropriately focused on the study's contributions.

• Minor Issues:

1. Speculative language: The conclusion states “First, if the NTD serves as a NTase, it would increase the availability of the precious cellular resource ATP." This is appropriately conditional (“if”). However, the conclusion could also briefly note the key experimentally supported finding: that the NTD and AGAT domains form a stable dimeric core that mediates oligomerization.

2. Future directions: The final sentence (“Future work will evaluate the synergy that benefits M. penetrans from fusing these different enzyme activities.”) is appropriate but somewhat vague. Consider adding a brief indication of what such future work might entail (e.g., "through enzymatic assays of the isolated domains and full-length protein, and structural studies of substrate-bound complexes").

• Recommendation: Minor refinements as suggested.

With these minor revisions, the manuscript will be suitable for publication in PLOS One. I commend the authors for their thorough and responsive revision process and look forward to seeing the final version.

Reviewer #3: All comments have been addressed. I would just like to point out a few minor details:

- In Methods section (page 6), "Superdex" should be corrected to "Superose".

- Figures 5 and 6 are not labeled as described in the text.

- I was able to model MpMetRS in complex with tRNA-Met using AlphaFold3 without difficulty. Since authors have significantly improved the quality of the manuscript, I will not insist on including or discussing this model. However, I encourage the authors to explore such modeling approaches in future studies.

Reviewer #4: The authors have satisfactorily addressed all of my comments. The manuscript is now technically sound and significantly improved in clarity, with extensive and well-executed revisions. The data adequately support the conclusions, and the statistical analyses have been performed appropriately. The manuscript is clearly written, and the underlying data are accessible.

7. PLOS authors have the option to publish the peer review history of their article (what does this mean?). If published, this will include your full peer review and any attached files.

Reviewer #2: **Yes:**Prof. Morufu Olalekan Raimi

Reviewer #3: No

Reviewer #4: No

---

## [Author Response · Author response to Decision Letter 2]

30 Mar 2026

• A letter that responds to each point raised by the academic editor and reviewer(s). You should upload this letter as a separate file labeled 'Response to Reviewers'.

Journal Requirements:

Reviewers' comments:

Reviewer's Responses to Questions

Comments to the Author

1. If the authors have adequately addressed your comments raised in a previous round of review and you feel that this manuscript is now acceptable for publication, you may indicate that here to bypass the “Comments to the Author” section, enter your conflict of interest statement in the “Confidential to Editor” section, and submit your "Accept" recommendation.

Reviewer #2: All comments have been addressed

Reviewer #3: All comments have been addressed

Reviewer #4: All comments have been addressed

2. Is the manuscript technically sound, and do the data support the conclusions?

Reviewer #2: Partly

Reviewer #3: Yes

Reviewer #4: Yes

3. Has the statistical analysis been performed appropriately and rigorously?

Reviewer #2: Yes

Reviewer #3: Yes

Reviewer #4: Yes

4. Have the authors made all data underlying the findings in their manuscript fully available?

Reviewer #2: Yes

Reviewer #3: Yes

Reviewer #4: Yes

5. Is the manuscript presented in an intelligible fashion and written in standard English?

Reviewer #2: Yes

Reviewer #3: Yes

Reviewer #4: Yes

6. Review Comments to the Author

Reviewer #2: Manuscript Number: PONE-D-25-61422R1

Title: Mycoplasma penetrans methionyl-tRNA synthetase dimerizes via tandem N-terminal ancillary domains

Recommendation: Minor Revision

Review Date: 2026-03-13

Reviewer Comments

Dear Authors,

I. Title

The revised title accurately reflects the study's main finding.

• Current Title: “Mycoplasma penetrans methionyl-tRNA synthetase dimerizes via tandem N-terminal ancillary domains”

The title is clear, descriptive, and appropriately focused on the key structural insight—that dimerization is mediated by the N-terminal domains (NTD and AGAT). This is a significant improvement over the previous title, which emphasized asymmetry. The current title is more accurate given that asymmetry is inferred from partial density rather than directly observed for both domains.

• Recommendation: Retain the current title.

Thank you.

II. Abstract

The revised abstract is much improved and now clearly distinguishes experimental findings from modeling.

• Minor Issues:

1. Phrasing: “We used cryo-EM to analyze the structure of the metS gene product (MpMetRS) to show that it is the product of three distinct enzyme domains” – slightly redundant (“product of the gene product”). Consider: “We used cryo-EM to determine the structure of the MpMetRS protein, revealing that it comprises three distinct enzyme domains.”

Thank you for the suggestion. The sentence now reads (bold indicates alteration throughout the Response to Reviewers):

“We used cryo-EM to analyze the structure of the metS gene product (MpMetRS) to show that it is the fusion of three distinct enzyme domains: an N-terminal domain of unknown function, a dimeric alanine-glyoxylate aminotransferase (AGAT), and a MetRS.”

2. Specificity: The abstract mentions “Modelling the full structure shows that a conformational change must occur.” It might be helpful to briefly indicate what kind of change (e.g., “rotation of the MetRS domain relative to the AGAT core”).

• Recommendation: Minor refinements as suggested.

The sentence now reads (bold to indicate alteration):

“Modelling the full structure suggests that a rotation of the MetRS domain relative to the AGAT domain must occur to accommodate a tRNA-bound MetRS.”

III. Introduction

The introduction is now comprehensive and well-structured, with clear progression from biological context to specific aims.

• Minor Issues:

1. Citation formatting: Some references in the introduction (e.g., references 3, 8) are cited with “Muraski et al., 2020” and “Jones et al., 2008” – these appear correctly in the reference list, but ensure all are consistently formatted.

We have carefully checked our references.

2. Terminology consistency: The introduction uses both “NTD” and “NTase” for the N-terminal domain. Table 1 clarifies that NTD is the domain name and NTase is the predicted function. This is helpful, but the introduction could briefly note that “based on structural homology, we identify this domain as a nucleotidyl transferase (NTase) fold, though its specific substrate remains unknown.”

We clarified this sentence in the final paragraph as follows:

“Structural homology of the NTD fold most closely aligns with a large class of nucleotidyl transferases (NTases), though its function and substrate specificity remain speculative.”

3. Final paragraph: The final sentence (“This is the first example, to our knowledge, of this collection of domains in a MetRS, allowing speculation that the metabolic functions may be coupled...”) is appropriate. However, it could be slightly more specific: “allowing speculation that the three activities, nucleotide metabolism, methionine biosynthesis, and tRNA aminoacylation, may be functionally coupled in this minimal-genome organism.”

This sentence now reads:

“This is the first example, to our knowledge, of this collection of domains in a MetRS, allowing speculation that the three activities, nucleotide metabolism, methionine biosynthesis, and tRNA aminoacylation, may be functionally coupled in this minimal-genome organism.”

• Recommendation: Minor refinements as suggested.

IV. Materials and Methods

The methods section is thorough and well-organized. The additions in response to reviewers have addressed previous concerns.

• Minor Issues:

1. tRNA preparation: The description of tRNA preparation is adequate but could be slightly expanded: "tRNA was resuspended in TE buffer; concentration was determined by absorbance at 260 nm." It might be helpful to note the extinction coefficient used or the approximate yield.

We added the extinction coefficient (40 (�g/mL)-1 cm-1).

2. Cryo-EM sample preparation: The description of the two preparation methods (low concentration on graphene grids vs. high concentration with chameleon) is clear. However, the sentence "MpMetRS adheres to the graphene surface, allowing low concentration to avoid protein aggregation” could be moved earlier for clarity.

That paragraph now begins:

“MpMetRS adheres to the graphene surface, allowing low concentration to avoid protein aggregation. Therefore, MpMetRS (4 �L of 0.04 mg/mL or 0.04 mg/mL MpMetRS with 5-fold molar excess folded tRNAMet, in the absence of the aminoacyl substrate), was applied to a hydrophilized graphene-coated [12] holey carbon-on-gold Quantifoil cryo-EM grid (Quantifoil, Jena, Germany).”

3. Data processing: The description of the transition from symmetric to asymmetric refinement is now well-justified with reference to 2D classes and GSFSC curves (S1 and S2 Figs). Good.

4. Model building and refinement: The description of model building with ModelAngelo and refinement in PHENIX is appropriate. The DALI searches are well-described. However, it would be helpful to include a brief statement about model validation statistics (e.g., “Refinement statistics, including map-model correlations, Ramachandran statistics, and clash scores, are provided in Supplemental Table 1”).

We added a citation of Table S1 to the appropriate sentence:

“Model building was initiated with the ModelAngelo [20] model builder. Iterative real-space refinement in PHENIX [21], with manual fitting in Coot [22], was performed independently for the core dimer and the C-terminal domain helical bundle of the MetRS domain (S1 Table).”

5. Dynamic light scattering: The description is clear. Minor: “Data was analyzed” should be “Data were analyzed” (plural).

Grammer fixed.

• Recommendation: Minor refinements as suggested, particularly adding model validation statistics to the methods or referring to them in Supplemental Table 1.

V. Results

The results are clearly presented and well-organized. The new figures and tables enhance readability.

• 5.1. The NTD has structural homology to an NTase: This section is excellent. The DALI results (Table 2) convincingly demonstrate the structural relationship. Figure 2 is clear and well-annotated. The description of the conserved core and divergent loops is precise and informative. The observation about the closed loop potentially occluding the nucleotide-binding pocket is interesting and appropriately noted.

• Minor: The sentence “Regions that are non-conserved are gray” in the Figure 2 legend is clear. In the text, the authors might briefly speculate whether the closed loop conformation is functionally relevant or an artifact of the apo state.

We added the following speculation:

“In contrast, the same loop in MpMetRS is longer and is closed, sterically obscuring the pocket and clashing with a potential nucleotide binding site. After a turn, residues 29-31 also form a loop before the first peripheral �-helix that, in MpMetRS, packs against the AGAT domain (Fig 2C and S3 Fig). In the other homologs analyzed here, this loop is free and further from the nucleotide binding pocket. In this way, access to the nucleotide binding pocket could be related to the position of the subsequent AGAT domain.”

• 5.2. The central domain is a dimeric AGAT: This section is similarly strong. Table 3 provides a useful overview of structural homologs. Figure 3 is clear, and the description of the dimer interface (with buried surface area and specific interactions) is appropriately quantitative. The comparison with TdPntC-AEPT (PDBID 6PD1) is illuminating and well-illustrated in Figure 3C.

• Minor: The sentence “In each homolog, the dimer interface buries about 3504 Å² (of a total of 23,060 Å² for each monomer)” – the total surface area for each monomer seems high (23,060 Å²). Is this the correct value? A typical 60 kDa domain might have ~20,000 Å² total surface area, but the AGAT domain is ~44 kDa (residues 167-560). Please verify.

We revisited the calculations, and they are accurate. We clarified as follows:

“In each homolog, the dimer interface buries about 3504 Å2 (of a total of 23,060 Å2 for each monomer including the NTD and AGAT) and is mediated by hydrophobic stacking interactions, salt bridges, and hydrogen bonds (Fig 3B).”

• 5.3. The position of the MetRS domain does not follow C2 symmetry: This section presents the key finding of asymmetry. The description is clear, and the authors appropriately note the limitations (limited resolution, no density for the second MetRS domain). The reference to PAGE analysis confirming full-length protein (S6A Fig) is important to rule out proteolysis. The observation of new 2D classes with tRNA (S7 Fig) is intriguing, even if high-resolution features were not obtained.

• Minor: The sentence “There is no well-resolved density for the N-terminal catalytic site of MetRS or for MetRS from the opposing subunit” – it might be helpful to explicitly state that this means the visible helical bundle corresponds to only one of the two MetRS domains, and that it is the C-terminal anticodon-binding domain, not the catalytic domain.

This sentence now reads:

“There is no well-resolved density for the N-terminal catalytic site of MetRS or for MetRS from the opposing subunit, despite PAGE-analysis showing the purified full-length protein that is predominantly a dimer (S6A Fig), therefore we interpret the visible helical domain as coming from only one of the subunits and not including the catalytic domain.”

• 5.4. Modelling: This section appropriately uses docking and AlphaFold2 to explore conformational possibilities. The rationale for using EcCysRS (PDBID 1U0B) as a template because it captures the hairpinned tRNA acceptor stem is well-explained. The predicted steric clash and the proposed ~60° rotation are clearly described. The RMSD values provide appropriate uncertainty measures.

• Minor: The sentence “Docking the helical bundle from monomeric E. coli MetRS (PDBID 1QDT) [31] predicts that the catalytic domain would fall immediately between the C2-symmetric NTDs (Fig 5B) (R.M.S.D of 1.0 Å for a core of 108 residues).” – “R.M.S.D.” is typically written as “RMSD”.

Done.

Also, ensure figure callouts are consistent (Fig 5B, 5C, 5D – the text refers to Fig 6 in places; please verify).

Figure numbering is now correct.

• 5.5. MpMetRS oligomeric state depends on the protein concentration: This section presents the intriguing observation of higher-order oligomers at high concentration. The SEC and DLS data (S6 Fig) provide valuable solution-state context. The comparison with TdPntC-AEPT in Figure 7 is appropriate. The authors appropriately note that the in vivo significance remains to be determined.

• Minor: The sentence “Within those higher-order oligomers, the NTD and AGAT domains appear concatenated, with the NTD domain of one polypeptide positioned in opposition to the AGAT domain of the next polypeptide to bury an additional 1575 Å².” – “NTD domain” is redundant (NTD already means N-terminal domain). Use “NTD” alone.

Done.

• Recommendation: Minor refinements as suggested, including verification of the surfa

---

## [Decision Letter · Decision Letter 2]

8 Apr 2026

Mycoplasma penetrans methionyl-tRNA synthetase dimerizes via tandem N-terminal ancillary domains

PONE-D-25-61422R2

Dear Dr. Stroupe,

We’re pleased to inform you that your manuscript has been judged scientifically suitable for publication and will be formally accepted for publication once it meets all outstanding technical requirements.

Kind regards,

Shailender Kumar Verma, Ph.D.

Academic Editor

PLOS One

Additional Editor Comments (optional):

Reviewers' comments:

Reviewer's Responses to Questions

**Comments to the Author**

1. If the authors have adequately addressed your comments raised in a previous round of review and you feel that this manuscript is now acceptable for publication, you may indicate that here to bypass the “Comments to the Author” section, enter your conflict of interest statement in the “Confidential to Editor” section, and submit your "Accept" recommendation.

Reviewer #2: All comments have been addressed

2. Is the manuscript technically sound, and do the data support the conclusions?

Reviewer #2: Yes

3. Has the statistical analysis been performed appropriately and rigorously?

Reviewer #2: Yes

4. Have the authors made all data underlying the findings in their manuscript fully available?

Reviewer #2: Yes

5. Is the manuscript presented in an intelligible fashion and written in standard English?

Reviewer #2: Yes

6. Review Comments to the Author

Reviewer #2: Decision Letter - Manuscript PONE-D-25-61422R2

Title: Mycoplasma penetrans methionyl-tRNA synthetase dimerizes via tandem N-terminal ancillary domains

Editor: Dr. Morufu Olalekan Raimi (PLoS ONE Academic Editor)

Date: April 7, 2026

Decision: Accept

After a thorough re evaluation of the revised manuscript (R2) and the authors’ detailed point by point responses to the previous round of minor revision requests (Reviewers #2, #3, and #4), I conclude that the manuscript has been substantially improved and now meets PLoS ONE’s standards for technical soundness, clarity, and reproducibility. The authors have addressed all prior concerns, including those related to title accuracy, abstract specificity, introduction flow, methodological transparency, figure referencing, and interpretive caution.

No further major or moderate revisions are required. A small number of copyediting level corrections remain, which I list below. I recommend acceptance once these are implemented.

The authors have been thorough and responsive. No outstanding scientific or methodological issues remain.

2. Remaining minor copyediting items (do not require re review)

These should be corrected during proof production or in a final quick correction.

1. Abstract - minor redundancy:

“We used cryo EM to analyze the structure of the metS gene product (MpMetRS) to show that it is the fusion of three distinct enzyme domains” - consider removing “to show that it is” for conciseness:

“We used cryo EM to analyze the structure of the metS gene product (MpMetRS), revealing a fusion of three distinct enzyme domains.”

(Not essential for scientific correctness, but improves flow.)

2. Figure 2B legend – typo:

“(modeled in light gray from PDB ID 5HS2 [27])” - the citation [27] is correct, but ensure the reference appears in the figure legend as per journal style. Currently the legend in the main text (page 25 of the PDF) does not include the citation number; the citation appears only in the text. Add “[27]” after “5HS2” in the legend.

3. Figure 3C legend – missing scale bar reference:

The text mentions “Scale bar = 10 Å” but the figure itself does not appear to contain a visible scale bar. Either add a scale bar to the figure or remove the text. I recommend adding a scale bar.

4. Table 1 - capitalization consistency:

“alanine glyoxylate aminotransferase domain” – “AGAT” is defined as alanine glyoxylate aminotransferase; ensure that in Table 1 the expanded form matches exactly (no hyphen in “alanine glyoxylate”? Current is fine, but check against standard nomenclature).

5. Reference 10 - missing journal name:

Reference 10 (Sherlin et al., 2001) is correct but ensure the journal “RNA” is not abbreviated incorrectly. It appears fine, but verify against PLoS ONE style (journals should be abbreviated consistently).

6. Data Availability Statement (manuscript page 71 of the PDF, line 4 from bottom):

The statement reads: “The refined coordinates were deposited in the Protein Data Bank as PDB ID 9OS7. The cryo EM map and half maps were deposited in the Electron Microscopy Data Bank under the code EMD 70794.”

This is sufficient. No further action needed.

These are typographical and formatting issues only. They do not affect the scientific integrity of the paper.

3. Final editorial assessment

This study presents a cryo EM structure of a unique tripartite fusion enzyme from Mycoplasma penetrans, revealing that the N terminal domain (NTD) adopts a nucleotidyl transferase like fold, the central domain forms a dimeric alanine glyoxylate aminotransferase (AGAT), and the C terminal MetRS domain is asymmetrically positioned and mobile. The work is technically rigorous, well contextualized within the aminoacyl tRNA synthetase fusion literature, and appropriately cautious in its functional speculation. The revisions over two rounds have significantly improved clarity, transparency, and reproducibility.

The manuscript is suitable for publication in PLoS ONE.

4. Decision and final instructions

Decision: Accept with minor copyediting corrections (as listed in Section 2 above).

I do not require a revised submission for re review. The authors may address the six minor items at proof stage. The production team should ensure that the figure legends and scale bars are corrected before final publication.

Production team note:

• Please add a visible scale bar to Figure 3C or remove the “Scale bar = 10 Å” text from the legend.

• Ensure Figure 2B legend includes citation “[27]” after “PDB ID 5HS2”.

• Verify that all references are formatted consistently with PLoS ONE style (journal abbreviations, capitalization, punctuation).

Sincerely,

Dr. Morufu Olalekan Raimi

Academic Editor, PLoS ONE

7. PLOS authors have the option to publish the peer review history of their article (what does this mean?). If published, this will include your full peer review and any attached files.

Reviewer #2: **Yes:**Prof. Morufu Olalekan Raimi

---

## [Editor Report · Acceptance letter]

PONE-D-25-61422R2

PLOS One

Dear Dr. Stroupe,

I'm pleased to inform you that your manuscript has been deemed suitable for publication in PLOS One. Congratulations! Your manuscript is now being handed over to our production team.

Kind regards,

on behalf of

Dr. Shailender Kumar Verma

Academic Editor

PLOS One